# VISA: Preserving Fine-Grained Perception in MLLMs via Visual Semantic Anchoring

## Abstract

Multimodal Large Language Models (MLLMs) have achieved remarkable success in general-purpose visual understanding. However, their training paradigm faces a fundamental bottleneck: the challenge of learning high-fidelity visual representations from indirect, text-based objectives alone. This inefficient process leads to a phenomenon we term semantic attenuation, where internal visual representations lose critical, high-fidelity details, hindering performance on tasks requiring fine-grained perception. To resolve this core representation learning challenge, we propose **VIsual Semantic Anchoring (VISA)**, a novel and general training framework that introduces a direct, vision-native supervisory signal into the MLLM's intermediate layers. By anchoring the MLLM's representations to the rich feature space of a pretrained Vision Foundation Model (VFM) through representation alignment, VISA ensures its visual pathway learns and maintains a detailed and structured understanding of the visual world. Our composite loss, which enforces both point-wise semantic alignment and structural consistency, makes this process highly effective. Extensive experiments on diverse benchmarks and model backbones demonstrate that by fostering more robust internal representations, VISA significantly enhances fine-grained reasoning, improves factual grounding against hallucinations, and accelerates training convergence, establishing a new and effective paradigm for developing more perceptually robust MLLMs. Our code is open-sourced via
https://anonymous.4open.science/r/anonymous_VISA-D482/

## 1 Introduction

The convergence of vision and language has marked a pivotal moment in artificial intelligence, spearheaded by the advent of Multimodal Large Language Models (MLLMs) (OpenAI, 2023; Team et al., 2023; Chen et al., 2024b; Schmidt & Chen, 2024). By integrating powerful vision encoders with sophisticated large language models (LLMs) through learned projection interfaces (Liu et al., 2023; Touvron et al., 2023; Chiang et al., 2023), these systems demonstrate a remarkable capacity for general-purpose visual reasoning. However, this success belies a fundamental training challenge. The predominant paradigm relies exclusively on a text-centric language modeling loss, which serves as an indirect and often inefficient proxy for learning high-fidelity visual representations. This forces the model to reshape complex visual information solely to serve the final text generation task, placing a substantial burden on the visual representation learning process itself.

This inefficiency directly leads to a critical deficiency: MLLMs often falter on tasks requiring fine-grained visual perception (Tong et al., 2024b; Qi et al., 2025; Ma et al., 2023), such as counting objects or discerning spatial arrangements (Yuksekgonul et al., 2022). We identify this failure as a direct symptom of semantic attenuation. From a representation learning perspective, this phenomenon occurs because visual details that lack direct, high-frequency textual correlates are progressively down-weighted, abstracted, or lost under the exclusive pressure of a text-only objective (Li & Zhang, 2024; Venhoff et al., 2025; Neo et al., 2024). This process creates a significant semantic gap between the MLLM's degraded internal features and the structured, high-fidelity representations captured by specialized Vision Foundation Models (VFMs).

To counteract this fundamental representation learning challenge, we introduce **VIsual Semantic Anchoring (VISA)**. Our framework is built on a simple yet powerful premise: to learn and preserve fine-grained visual detail, an MLLM's visual pathway requires a direct, vision-native supervisory

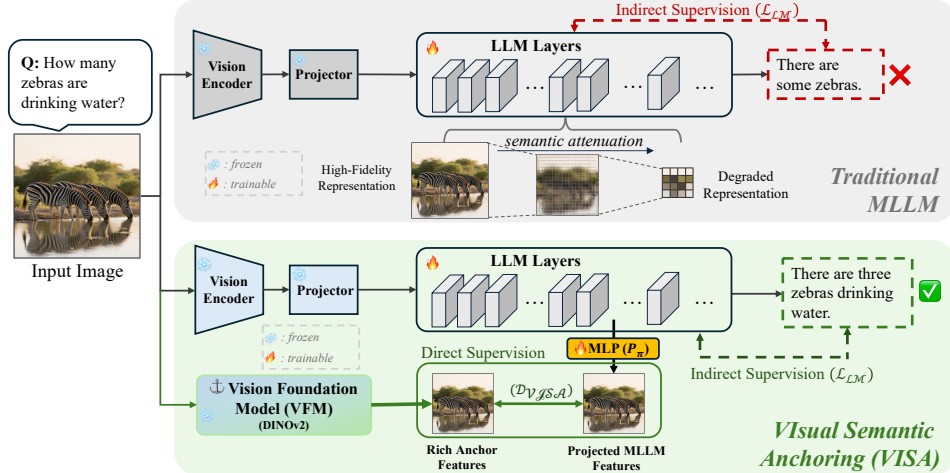

Figure 1: A comparison between traditional MLLM training and our proposed VISA framework. **Top:** In traditional MLLMs, indirect supervision from a text-only objective leads to semantic attenuation, where fine-grained visual details are lost. **Bottom:** VISA introduces direct, vision-native supervision by anchoring the MLLM's internal representations to the rich feature space of a Vision Foundation Model (VFM).

signal, rather than an indirect textual one. VISA provides this by leveraging a pretrained, vision-specialized VFM as an unwavering semantic anchor. We specifically employ a VFM trained via self-supervision on images alone, such as DINOv2 (Oquab et al., 2023; Kirillov et al., 2023; Yang et al., 2024; Ranzinger et al., 2024). Its feature space, optimized purely for visual semantics, lacks the linguistic alignment bias inherent in models like CLIP (Radford et al., 2021b), which are pretrained on image-text pairs. These vision-native representations are therefore less susceptible to semantic attenuation, providing an ideal anchor for high-fidelity visual structure.

Methodologically, VISA operates as a form of feature-level knowledge distillation (Hinton et al., 2015; Romero et al., 2015; Gou et al., 2021), enabling the MLLM to implicitly ensemble diverse visual experts. Unlike recent multi-encoder frameworks (Kar et al., 2024; Cao et al., 2025; Shi et al., 2024; Tong et al., 2024a) that explicitly concatenate features at the expense of increased inference latency, VISA internalizes the anchor's fine-grained percepts during training only. This strategy grants the model the perceptual robustness of dual-encoder systems with zero additional inference overhead, effectively resolving the trade-off between visual breadth and computational efficiency. By establishing this stable, vision-native reference, VISA implements its guidance through a targeted auxiliary loss that performs representation alignment (Yu et al., 2025). This creates a continuous supervisory signal from the anchor model to the MLLM's internal processing layers. The signal encourages the MLLM's intermediate visual representations to maintain a detailed view of the visual world. This direct supervision simplifies the model's overall learning task by decoupling the objectives of visual fidelity and linguistic fluency, fostering a richer multimodal alignment. Our contributions are summarized as follow:

- We identify and validate a key limitation for general MLLM training from semantic attenuation perspective, where text-only supervision degrades the integrity of internal visual representations.
- We propose VISA, a novel and general training framework that resolves inefficiency of learning visual representations from indirect, text-only supervision by leveraging a VFM as a semantic anchor to provide direct, fine-grained supervision to the MLLM's internal visual pathway.
- We conduct a comprehensive suite of experiments across multiple modern backbones and challenging benchmarks, confirming that VISA delivers substantial improvements in visual acuity, training efficiency, and overall robustness.

## 2 RELATED WORK

**The MLLM Architecture and its Input Gateway.** The standard architecture of contemporary MLLMs is a tripartite system, composing a pretrained vision encoder, a lightweight projector, and a

large language model core. A significant body of research aimed at enhancing their visual acuity has focused on this initial input gateway. One major thrust involves leveraging more powerful, specialized vision encoders to supply the language model with a richer and more discriminative set of visual features from the outset (Kar et al., 2024; Lu et al., 2024; Shi et al., 2024; Azadani et al., 2025). Complementary to this, another line of work has sought to improve the computational efficiency of this gateway, developing sophisticated token pruning and merging strategies to reduce the number of visual tokens processed by the LLM without substantial performance degradation (Vasu et al., 2025; Yang et al., 2025; Wen et al., 2025; Tanaka & Watanabe, 2025). These approaches have collectively advanced the quality and economy of the visual information available to MLLMs.

**Internal Dynamics and Reconstruction-based Supervision.** A growing body of analytical work has begun to demystify the internal journey of visual information within the language model's deep architecture. These studies reveal a consistent pattern of functional specialization across layers: early layers tend to perform a coarse aggregation of the global visual scene, while the final layers are primarily concerned with integrating multimodal concepts for coherent text generation (Kaduri et al., 2025; Zhang et al., 2025; Choi & Park, 2025; Kimura & Ito, 2024). To address information loss within these layers, reconstruction-based methods have been proposed. For instance, X-Former (Swetha et al., 2024) and ROSS (Wang et al., 2024) employ auxiliary objectives to reconstruct input images or raw features from internal states. While these methods share our motivation of preserving detail, they typically enforce low-level fidelity (e.g., pixel reconstruction) or align with the model's own degraded features. In contrast, VISA anchors representations to the high-level semantic feature space of a strong VFM, effectively prioritizing semantic consistency.

**Visual Representation Distillation.** Our approach fundamentally aligns with the paradigm of Feature Knowledge Distillation (Hinton et al., 2015; Romero et al., 2015), where a student model learns to mimic the intermediate representations of a stronger teacher. In the context of MLLMs, recent works such as BRAVE (Kar et al., 2024) and MoVE-KD (Cao et al., 2025) utilize distillation to combine features from multiple diverse encoders. However, these explicit ensemble approaches typically require running multiple encoders during inference, increasing computational overhead. VISA adopts a streamlined, implicit ensemble strategy: we distill the fine-grained perceptual capabilities of a VFM into the MLLM's single visual pathway during training only. This avoids complex attention-based distillation overheads (Wang et al., 2024) while retaining the efficiency of a single-encoder architecture at inference time.

## 3 METHODOLOGY

### 3.1 PRELIMINARIES

Multimodal Large Language Models (MLLMs) are designed to jointly process and reason about information from different modalities, primarily vision and language. The now-standard architecture, popularized by models like LLaVA and its successors (Liu et al., 2023; 2024a; Li et al., 2024), comprises three fundamental components: a pretrained vision encoder, a vision-language projector, and a large language model. In this setup, the vision encoder acts as the model's perceptual system, the projector serves as a cross-modal translator, and the LLM functions as the central reasoning engine that integrates information from both sources to perform complex tasks.

The process begins with the Vision Encoder, $V_\psi(\cdot)$, which is responsible for converting a raw input image $I \in \mathbb{R}^{H \times W \times C}$ into a sequence of high-level feature representations. Typically, this role is filled by a Vision Transformer (ViT) that has been pretrained on massive image or image-text datasets. A common choice is the CLIP ViT encoder (Radford et al., 2021a), which learns powerful and semantically rich visual features through large-scale contrastive training between images and their corresponding captions. The encoder processes the image by dividing it into a grid of patches and embedding each patch into a vector. The output is a sequence of $N$ visual feature vectors $\mathbf{z} = V_\psi(I) = \{z_1, z_2, ..., z_N\}$, where $\mathbf{z} \in \mathbb{R}^{N \times D_z}$ and $D_z$ is the feature dimension of the encoder. To preserve its robust, pre-learned perceptual capabilities, the vision encoder's parameters $\psi$ are typically kept frozen throughout the MLLM's instruction tuning phases.

Next, the Vision-Language Projector, $P_\phi(\cdot)$, serves as a crucial bridge between the distinct feature spaces of the vision encoder and the language model. This component is usually a lightweight, learnable module, such as a Multi-Layer Perceptron (MLP), that is designed to be both efficient and effective. Its function is to take the sequence of visual features $\mathbf{z}$ from the vision encoder and

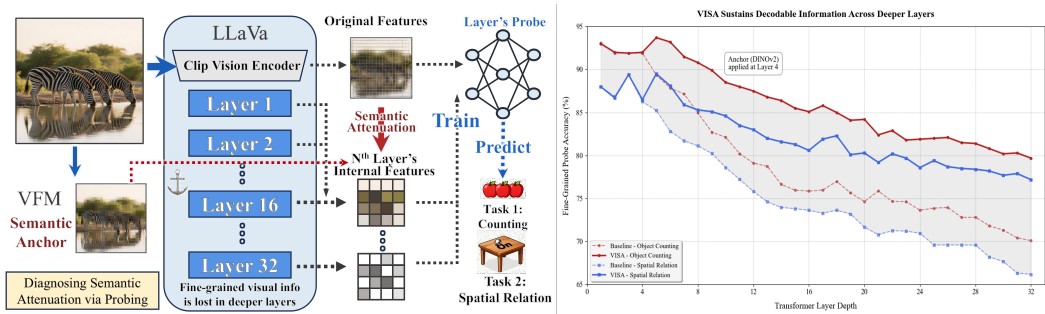

Figure 2: Layer-wise probe accuracy for fine-grained visual tasks. The baseline model (dashed lines) shows a rapid degradation of decodable information as layer depth increases. The VISA-enhanced model (solid lines) consistently maintains higher probe accuracy across all layers, demonstrating its effectiveness in mitigating information attenuation.

transform it into a new sequence of embeddings, $\mathbf{e}^{\text{img}} = P_\phi(\mathbf{z})$, where $\mathbf{e}^{\text{img}} \in \mathbb{R}^{N \times D}$. Here, $D$ is the hidden dimension of the LLM, making the visual tokens dimensionally compatible and semantically aligned with the LLM's word embedding space. The parameters $\phi$ of the projector are the primary focus of the initial vision-language alignment pretraining stage (Liu et al., 2023).

Finally, the LLM, $LM_\theta(\cdot)$, performs multimodal processing by taking the sequence of projected visual tokens $\mathbf{e}^{\text{img}}$ and prepending it to the sequence of standard text embeddings $\mathbf{e}^{\text{text}}$. The model then processes this concatenated sequence auto-regressively. The entire system is optimized using a standard Language Modeling Objective, which trains the model to predict the next text token given the preceding text and the visual context. The training loss, $\mathcal{L}_{\text{LM}}$, is the negative log-likelihood of the ground-truth text sequence $T = \{t_1, ..., t_K\}$:

$$\mathcal{L}_{\text{LM}} = -\sum_{i=1}^{K} \log p_{\theta,\phi}(t_i \mid t_{<i}, \mathbf{e}^{\text{img}}) \tag{1}$$

This objective is used across the two primary training phases: an initial alignment pretraining stage where only the projector parameters $\phi$ are updated, and a subsequent visual instruction tuning stage where both $\phi$ and the LLM parameters $\theta$ (often via low-rank adaptation, LoRA (Hu et al., 2022)) are fine-tuned to handle complex, instruction-based tasks (Liu et al., 2024a). While effective for language-centric tasks, this training objective provides only an indirect supervisory signal to the model's internal visual pathway, the consequences of which we analyze next.

### 3.2 MOTIVATION: DIAGNOSING THE SEMANTIC GAP VIA REPRESENTATION PROBING

As established, the MLLM's visual pathway is optimized via a distal, text-centric objective. This raises a question regarding the quality of the resulting internal representations: to what extent do they retain the fine-grained, separable visual information present in the original encoder features? To investigate this, we designed a series of diagnostic probing experiments to quantify the semantic gap between the MLLM's internal features and the high-fidelity source representations (detailed protocols on probing dataset construction and training settings are provided in Appendix D). A probe is a lightweight classifier, in our case a 2-layer MLP, trained to predict specific visual attributes directly from a set of frozen feature representations. High probe accuracy suggests that an attribute is easily decodable, whereas low accuracy indicates the information has been obfuscated or lost, signifying a wider semantic gap. We designed two distinct, fine-grained probing tasks: Object Counting and Spatial Relation classification, which are two common tasks in probing MLLM's general capabilities. We then trained these probes on features extracted from the original CLIP vision encoder and from the layers of a LLaVA-1.5-7B model.

The results, presented in Figure 2, reveal a stark trend. For both tasks, there is a clear decline in probe performance as we move deeper into the MLLM's architecture, exposing a widening semantic gap. This provides strong empirical evidence for our central hypothesis: the MLLM's internal visual representations undergo a form of semantic attenuation as a direct consequence of inefficient representation learning from a text-only objective. This observation aligns with recent

theoretical findings (Park et al., 2023) that supervision from high-level semantic tasks tends to induce homogeneity in deeper layers, effectively acting as a low-pass filter that sacrifices high-frequency details needed for fine-grained perception.

Furthermore, we observe that this loss of detail is linked to the optimization process. By tracking the High-Frequency Ratio (HFR) of visual features at Layer 16 throughout the SFT phase, we found that the baseline model exhibits a decline in high-frequency energy as training iterations increase (Figure 3). This indicates that the text-only objective compels the model to forget fine-grained visual details. Please refer to appendix E for more detail.

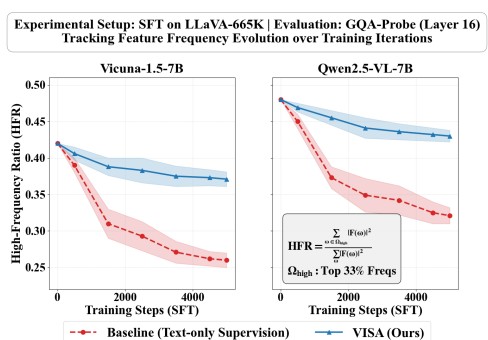

In the process of adapting to the language modeling objective, the model transforms its visual features in a way that entangles and abstracts concepts, making the original, disentangled attributes significantly harder to decode. This finding directly motivates the development of a mechanism to actively bridge this representation gap, which we introduce next.

Figure 3: **Evolution of High-Frequency Information during SFT.** The High-Frequency Ratio (HFR) of the baseline model (Red) declines progressively, indicating a loss of detail. VISA (Blue) effectively preserves these fine-grained signals throughout training.

### 3.3 VISA: Representation Alignment via Composite Semantic Anchoring

To bridge the empirically observed semantic gap, we introduce **VIsual Semantic Anchoring (VISA)**, a training framework designed to guide the MLLM's internal representations through representation alignment with the rich, stable feature space of a powerful Vision Foundation Model (VFM). The VFM serves as a semantic anchor, providing a consistent, high-quality target representation for the MLLM's visual pathway. This alignment process is mediated by a learnable semantic adapter and optimized via a novel, composite loss function that preserves both token-level semantics and scene-level structure.

**Formal Definitions.** Let $I$ be an input image. The frozen VFM encoder, $\mathcal{E}(\cdot)$, which acts as our semantic anchor, maps the image to a set of $N$ target feature vectors $Y = \{y_1, \ldots, y_N\}$, where $Y \in \mathbb{R}^{N \times d}$, where $d$ is the embedding dimension of $\mathcal{E}(\cdot)$. Concurrently, within the MLLM, the visual tokens at a chosen intermediate layer $\ell$ are denoted by $E_\ell = \{e_{\ell,1}, \ldots, e_{\ell,N}\}$, where $E_\ell \in \mathbb{R}^{N \times D}$ and $D$ is the embedding size of MLLM's vision encoder. To bridge the two potentially different feature spaces, we introduce a learnable, lightweight MLP as a semantic adapter, $P_\pi : \mathbb{R}^D \to \mathbb{R}^d$. This adapter projects the MLLM's internal representations into the VFM's feature space, yielding the adapted representations $E'_\ell = P_\pi(E_\ell)$ to be aligned. We implement $P_\pi$ as a 3-layer MLP to ensure sufficient capacity for feature mapping (see Appendix C.1 for architectural details). In cases where the VFM's resolution differs from the MLLM's, resulting in mismatched token counts, we apply bilinear interpolation to the VFM's feature map to spatially align $Y$ with $E_\ell$ before computing the loss.

Our core contribution is a composite alignment strategy, materialized through two complementary loss components that together quantify the divergence from the ideal visual representation provided by the anchor.

**Weighted Point-wise Semantic Loss.** This objective enforces semantic alignment between the adapted MLLM and VFM at the individual token level, with a strategic focus on the most salient regions of the image. For self-attention based VFMs such as DINOv2 (Oquab et al., 2023), which utilize a class [CLS] token for global image representation, a natural saliency map can be derived. These scores are then normalized via a Softmax function over the VFM's final layer attention from the [CLS] token to all patch tokens, yielding the weight vector $w = \text{Softmax}(\{A_{[\text{CLS}],i}\}_{i=1}^N)$. The loss is then formulated as the weighted negative cosine similarity between the adapted MLLM tokens and the VFM target tokens:

$$\mathcal{L}_{\text{point}} = -\sum_{i=1}^{N} w_i \cdot \frac{E'_{\ell,i} \cdot y_i}{\|E'_{\ell,i}\|_2 \cdot \|y_i\|_2} \tag{2}$$

This loss term quantifies the semantic discrepancy at a fine-grained, token-wise level.

**Structural Consistency Loss.** Beyond aligning individual token semantics, preserving the geometric structure of the representation space is equally critical. To achieve this, our second objective enforces structural consistency between the two feature spaces. We capture the internal structure of a feature set $F$ by computing its Gram matrix, $G(F) = \hat{F}\hat{F}^T$, where $\hat{F}$ represents the L2-normalized features. The Gram matrix encodes the complete set of pairwise similarities between all tokens in the sequence. We then minimize the distance between the Gram matrix of the VFM's target features and that of the MLLM's adapted features:

$$\mathcal{L}_{\text{struct}} = \frac{1}{N^2}\|G(Y) - G(E'_\ell)\|_F^2 \tag{3}$$

where $\|\cdot\|_F$ is the Frobenius norm. This loss term captures the holistic, relational distortion between the two representation spaces.

**Overall Training Objective.** We unify these two components into a single, tunable metric we term the **Anchoring Divergence**, $\mathcal{D}_{\text{VISA}}$. This metric provides a holistic measure of how far the MLLM's internal representation has drifted from the VFM's stable anchor. It is defined as a weighted combination of the point-wise and structural losses, with the hyperparameter $\alpha$ controlling the balance between them:

$$\mathcal{D}_{\text{VISA}}(E'_\ell, Y|\alpha) = \mathcal{L}_{\text{point}} + \alpha \cdot \mathcal{L}_{\text{struct}} \tag{4}$$

This divergence is then integrated into the MLLM's primary training objective as a regularization term. The final objective function is a composite of the standard language modeling loss and our Anchoring Divergence, where the hyperparameter $\lambda$ governs the overall strength of the alignment signal:

$$\mathcal{L}_{\text{total}} = \mathcal{L}_{\text{LM}} + \lambda \cdot \mathcal{D}_{\text{VISA}}(E'_\ell, Y|\alpha) \tag{5}$$

By minimizing this composite objective, the MLLM is trained not only to be a fluent language generator but also to maintain an internal visual representation that is semantically detailed and structurally coherent with an expert visual model, directly addressing the semantic attenuation issue.

## 4 EXPERIMENTS

### 4.1 EXPERIMENTAL SETTINGS

**Baselines and Controlled Implementation.** To rigorously evaluate the effectiveness of VISA in isolation, we adopt a **controlled experimental setting**. We utilize the state-of-the-art MLLMs as initialization: Vicuna-1.5 (7B/13B) (Chiang et al., 2023), Qwen2.5-VL-7B (Bai et al., 2025), and InternVL3.5-8B (Wang et al., 2025). To ensure a fair comparison and disentangle the benefits of our method from the scale of training data, all models (both Baselines and VISA-enhanced versions) are Supervised Fine-Tuned (SFT) using **exclusively the LLaVA-665K dataset** (Liu et al., 2024a). Detailed hyperparameters and infrastructure settings are provided in Appendix A.

**VISA Framework Details.** For our VISA framework, we employ DINOv2 (ViT-L/14) (Oquab et al., 2023) as the default semantic anchor. The anchoring loss is applied to the 16th layer of the MLLM's visual pathway. The semantic adapter ($P_\pi$) is a lightweight 3-layer MLP. Unless specified otherwise, we set $\alpha = 0.5$ and $\lambda = 1.0$.

**Evaluation Benchmarks and Metrics.** We evaluate on a comprehensive suite of benchmarks covering fine-grained perception, reasoning, and general understanding:

- **Perception & VQA & Reasoning:** We use **MMVP** (Tong et al., 2024c) for visual patterns that CLIP struggles with. We evaluate on **GQA** (Hudson & Manning, 2019) for compositional reasoning, **TextVQA** (Singh et al., 2019) for OCR, and the challenging **MMMU(Val** (Yue et al., 2024) for multi-discipline reasoning.

- **Hallucination & General Understanding :** We report results on **MMBench** (Liu et al., 2024b) for robust evaluation, **MME** (Yin et al., 2023), **MM-Star** (Chen et al., 2024a), and **POPE** (Li et al., 2023) for object hallucination.

## 4.2 MAIN RESULTS

The quantitative results are summarized in Table 1. VISA delivers consistent and significant improvements across all backbones. On tasks demanding precise visual details, VISA achieves the most notable gains. For example, on MMVP, VISA boosts Vicuna-1.5-13B by **+6.4%** and Qwen2.5-VL-7B by **+3.1%**. VISA also enhances complex reasoning and General Understanding. On MMMU, MME, MMBench and MM-Star, VISA-enhanced models consistently surpass their baselines, indicating that better visual grounding translates to superior reasoning.

Table 1: Comprehensive performance comparison. All models are fine-tuned on LLaVA-665K. VISA consistently improves performance across diverse backbones. The best results for each setting are in **bold**.

| Base Model | Method | Perception & VQA & Reasoning | | | | Hallucination & General Understanding | | | |
| --- | --- | --- | --- | --- | --- | --- | --- | --- | --- |
| | | GQA | TextVQA | MMVP | MMMU | POPE | MME | MMBench | MM-Star |
| Vicuna-1.5-7B | Baseline | 61.2 | 57.6 | 28.0 | 34.5 | 85.1 | 1646 | 64.3 | 33.5 |
| | + VISA | **64.2** | **61.8** | **32.9** | **36.2** | **88.2** | **1691** | **66.8** | **37.1** |
| Vicuna-1.5-13B | Baseline | 62.8 | 60.9 | 38.9 | 36.8 | 87.8 | 1595 | 67.1 | 34.2 |
| | + VISA | **66.2** | **63.9** | **45.3** | **38.4** | **88.9** | **1631** | **69.5** | **37.4** |
| Qwen2.5-VL-7B | Baseline | 64.4 | 83.8 | 47.1 | 53.6 | 85.9 | 2254 | 72.4 | 61.6 |
| | + VISA | **67.1** | **85.7** | **50.8** | **57.2** | **86.5** | **2301** | **74.1** | **62.8** |
| InternVL3.5-8B | Baseline | 65.2 | 77.6 | 49.8 | 63.1 | 88.1 | 2369 | 79.8 | 68.8 |
| | + VISA | **68.5** | **80.3** | **52.3** | **64.2** | **88.7** | **2381** | **81.9** | **69.7** |

## 4.3 ABLATION STUDY

Having established the overall efficacy of VISA, we now dissect the framework to validate its core design principles. We structure our analysis as an investigation into what constitutes an effective representation alignment strategy, focusing on three fundamental questions: (1) Where in the MLLM's architecture is the alignment guidance most impactful? (2) What properties must the anchor's target representation possess? and (3) How does VISA compare to explicit multi-encoder ensembles in terms of efficiency?

### 4.3.1 INVESTIGATING THE OPTIMAL LOCUS FOR ANCHORING

We begin by identifying the most effective point of intervention for our semantic anchor. As prior work suggests, the intermediate layers of an MLLM are the primary locus for fine-grained multimodal fusion (Jiang et al., 2025; Kang et al., 2025). We systematically applied the VISA loss (using DINOv2 as the anchor) to different layers of the LLaVA-1.5-7B backbone. The results, visualized in Figure 4, exhibit a distinct unimodal trend, peaking sharply when the anchor is applied to the 16th layer. This confirms that the architectural midpoint is the most receptive locus for injecting vision-native guidance, balancing the preservation of low-level visual details with high-level semantic abstraction.

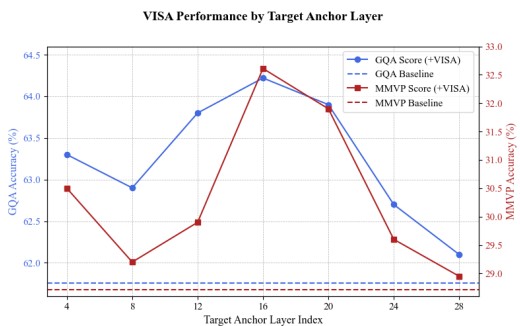

Figure 4: Performance on GQA and MMVP benchmarks as a function of VISA's target layer in LLaVA-1.5-7B.

### 4.3.2 DISSECTING THE ANCHOR AND LOSS DESIGN

We conducted comprehensive experiments, summarized in Table 2, to parse the contributions of the anchor's knowledge source and the specific components of our composite loss function.

To address the question of whether stronger contrastive models or generative models could serve as better anchors, we expanded our comparison to include **SigLIP** (Zhai et al., 2023) and **MAE** (He et al., 2021), as shown in Table 2. **SigLIP** outperforms CLIP-Vision as an anchor due to its stronger semantic representations, but it still falls short of DINOv2 on fine-grained tasks like MMVP. **MAE**

Table 2: Ablation studies on the anchor's knowledge source, loss components, and the importance of Saliency Weighting.Best results in **bold**.

| Ablation Group | Configuration | Perception, VQA & Reasoning | | | | Hallucination | General Understanding | | |
| --- | --- | --- | --- | --- | --- | --- | --- | --- | --- |
| | | GQA | TextVQA | MMVP | MMMU | POPE | MME | MMB | MM-Star |
| **Baseline** | | 61.2 | 57.6 | 28.0 | 34.5 | 85.1 | 1646 | 64.3 | 33.5 |
| **Anchor Source** | Self-Anchor (CLIP-Vision) | 62.5 | 58.9 | 29.8 | 35.1 | 87.1 | 1660 | 65.2 | 35.0 |
| | Language-Aligned (CLIP-Text) | 59.5 | 55.8 | 25.1 | 33.8 | 84.5 | 1590 | 63.1 | 31.8 |
| | Contrastive SOTA (SigLIP) | 63.9 | 60.4 | 31.2 | 35.9 | **88.4** | 1676 | 66.4 | 36.2 |
| | Reconstruction (MAE) | 62.3 | 59.2 | 31.0 | 34.9 | 86.8 | 1655 | 64.9 | 34.8 |
| | **Vision-Native (DINOv2)** | **64.2** | **61.8** | **32.9** | **36.2** | 88.2 | **1691** | **66.8** | **37.1** |
| **Loss Components** | VISA w/o Saliency Weighting | 63.6 | 60.5 | 31.7 | 35.7 | 87.6 | 1680 | 66.1 | 36.2 |
| | Structural Loss Only ($\mathcal{L}_{\text{struct}}$) | 63.2 | 60.2 | 31.8 | 35.5 | 87.5 | 1678 | 65.9 | 36.0 |
| | **Full VISA (Composite)** | **64.2** | **61.8** | **32.9** | **36.2** | 88.2 | **1691** | **66.8** | **37.1** |

provides strong structural guidance, improving localization, but lacks the high-level semantic abstraction needed for complex reasoning, confirming that pure pixel-level dependencies are insufficient. **DINOv2** achieves the best balance, offering both the rich semantic structure of a VFM and the fine-grained detail preservation of self-supervised learning. Moreover, We further ablate the Saliency Weighting mechanism in our point-wise loss (Eq. 2). Removing the weighting term leads to a noticeable drop in performance. This confirms that naively aligning all tokens introduces noise from background redundancy, whereas weighting the alignment based on the VFM's [CLS] attention effectively focuses the student model on foreground objects.

## 4.4 FURTHER ANALYSIS ON INTERNAL MECHANISMS AND EFFICIENCY

Having demonstrated VISA's effectiveness, we now provide a deeper analysis of how it alters the model's internal processing and enhances its practical training efficiency.

**Internal Representation Quality.** Our central hypothesis is that VISA improves performance by mitigating semantic attenuation. A frequency-domain analysis of the internal representations from Layer 16 provides strong evidence for this. As shown in Figure 5, the log power spectrum of the baseline model's features is heavily concentrated at the center, indicating a strong bias towards low-frequency components and a significant loss of high-frequency information. In contrast, the spectrum from the VISA-enhanced model displays a much broader energy distribution, with substantially more power in the outer regions corresponding to high-frequency signals. This demonstrates that by providing a direct, vision-native guidance signal, VISA effectively compels the MLLM to preserve the fine-grained, high-frequency details that are essential for a precise and rich visual representation.

**Practicality and Training Dynamics.** Beyond improving internal representations, VISA is also a practical and efficient training framework. To ensure our method is not reliant on brittle settings, we conducted a sensitivity analysis for its primary hyperparameters: the overall anchoring strength $\lambda$ and the structural loss weight $\alpha$. As shown in Figure 6a-b, VISA exhibits remarkable stability, maintaining high performance across a practical range of values for both parameters around our defaults ($\lambda = 1.0, \alpha = 0.5$). This indicates that VISA is a reliable enhancement that can be applied without exhaustive hyperparameter sweeps. Finally, we assessed VISA's impact on training convergence by tracking performance throughout the instruction tuning phase. The results, presented in Figure 6c, show that the direct guidance from the semantic anchor not only leads to a higher performance ceiling but also dramatically accelerates convergence for both GQA and MMVP. The VISA-enhanced model consistently outperforms the baseline at every checkpoint, suggesting that semantic anchoring provides a more efficient and effective learning signal than the distal text-only objective alone.

## 5 INSIGHTS AND DISCUSSION

**Why is a VFM a uniquely suitable semantic anchor?** MLLM training suffers from semantic attenuation, where text-centric optimization degrades fine-grained visual details not directly correlated with language. To counteract this, we anchor internal representations to a self-supervised Vision Foundation Model (VFM) like DINOv2. Its feature space, optimized purely for visual semantics, provides a stable, high-fidelity supervisory signal free from the linguistic bias inherent in models like CLIP. This vision-native guidance compels the MLLM to maintain a structured visual representation, directly

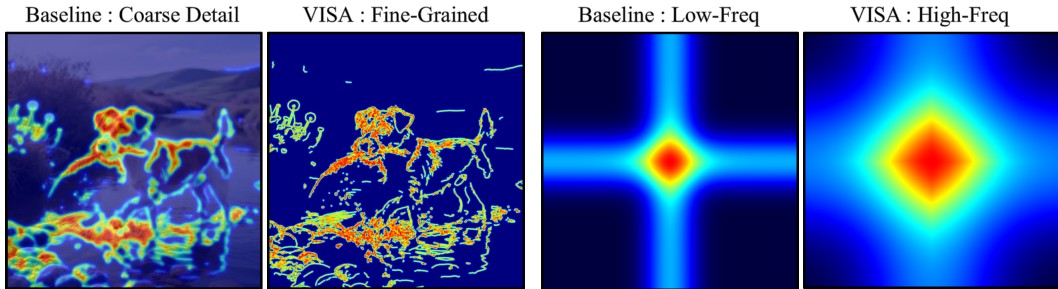

Figure 5: Frequency domain analysis of internal visual representations from Layer 16. The log power spectrum of the baseline model's features (center) is heavily biased towards low frequencies (center of the spectrum). The VISA-enhanced model (right) preserves significantly more high-frequency information (outer regions), indicating a richer, more detailed internal representation of the visual input.

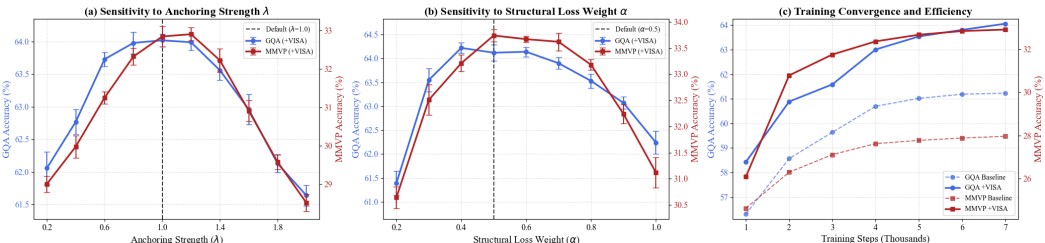

Figure 6: Hyperparameter sensitivity analysis on the GQA and MMVP benchmarks. (a) Performance variation with respect to the overall anchoring strength $\lambda$. (b) Performance variation with respect to the structural loss weight $\alpha$. The framework shows robust performance around our default values.

addressing the observed weakness of even top-tier MFMs in fine-grained geometric reasoning—a domain where they lag significantly behind vision specialists (Ramachandran et al., 2025).

**Why is anchoring at the architectural midpoint most effective?** Our results identify the architectural midpoint as the optimal locus for anchoring. A growing body of work confirms this functional specialization: early layers handle coarse visual grounding, while late layers are dominated by language-centric abstraction, making them less suitable for preserving fine-grained visual detail (Kaduri et al., 2025; Zhang et al., 2025). The midpoint thus represents a unique juncture where visual features are both semantically mature and retain maximal detail, making them most receptive to our vision-native guidance. Intervening at this stage directly counteracts the degradation of visual information that leads to object hallucination, a phenomenon prominently linked to processing deficits in these middle layers (Jiang et al., 2025). Applying the anchor here preserves the visual integrity required to prevent failures in complex, nonlocal reasoning tasks (Berman & Deng, 2025).

## 6 CONCLUSION

In this work, we identify and address semantic attenuation: a fundamental challenge where the indirect, text-only supervision in modern MLLMs leads to the degradation of fine-grained visual details in their internal representations. To resolve this, we introduce **VIsual Semantic Anchoring (VISA)**, a training framework that provides direct, vision-native supervision by anchoring the MLLM's intermediate representations to the rich feature space of a Vision Foundation Model. Our comprehensive experiments demonstrate that VISA consistently delivers substantial performance gains on challenging fine-grained VQA and hallucination benchmarks across multiple MLLM architectures. Furthermore, our analysis confirms that this approach not only enhances final accuracy but also improves internal representational quality and accelerates training convergence. Ultimately, VISA indicates that preserving high-fidelity visual grammar through semantic anchoring is a powerful paradigm for building more robust and perceptually keen multimodal AI systems.

ETHICS STATEMENT

The work presented in this paper is methodological in nature, focusing on the development of MLLM. To the best of our knowledge, our proposed methods do not introduce any new ethical concerns.

REPRODUCIBILITY STATEMENT

To facilitate the verification of our results, the implementation code for our algorithm and the main baselines is provided in the anonymous code link and the appendix.

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

## USE OF LARGE LANGUAGE MODELS

We utilized a large language model to enhance the language and clarity of our manuscript. Specifically, we employed Gemini 2.5 flash with the following prompt to refine the initial draft: *I am writing an academic paper in English. Please polish the following draft so that it adheres to the conventions of academic writing.*

## APPENDIX

## A EXPERIMENTAL SETUP DETAILS

This section provides a detailed account of our experimental configurations, including the hardware platform, model architectures, base codebase, and training hyperparameters, to ensure the full reproducibility of our research.

### A.1 HARDWARE PLATFORM

**Hardware Platform.** All experiments are run on 8 NVIDIA A100 GPUs, interconnected via NVLink for high-speed inter-GPU communication. We employed a Fully Sharded Data Parallel (FSDP) strategy, managed via the Accelerate library, to efficiently distribute the model parameters and optimizer states across all GPUs.

### A.2 MODEL CONFIGURATIONS

**Baseline Models.** We selected three state-of-the-art open-source MLLMs as our baselines. Their specific configurations are as follows:

- **LLaVA-1.5 (Vicuna):** Based on the Vicuna-v1.5 LLM (Chiang et al., 2023), we utilized both the 7B and 13B parameter versions (`lmsys/vicuna-7b-v1.5` and `lmsys/vicuna-13b-v1.5`).

- **Qwen2.5-VL-7B:** A high-performance MLLM from Alibaba (Bai et al., 2025). We used the `Qwen/Qwen2.5-7B-VL` version.

- **InternVL3.5-8B:** A powerful model known for its strong performance on a wide array of benchmarks (Wang et al., 2025). We used the `OpenGVLab/InternVL3.5-8B` version.

**VISA Framework Components.**

- **Semantic Anchor (VFM):** We use DINOv2 (ViT-L/14) (Oquab et al., 2023) as our default semantic anchor, specifically loading the `facebook/dinov2-large` model. We extract features from the final transformer block, using the patch token representations as the target feature set $Y$. The choice of DINOv2 is motivated by its strong performance on dense visual tasks, indicating its features are rich in spatial and structural information.

- **Semantic Adapter ($P_\pi$):** This adapter is a 3-layer MLP with a `Linear -> GELU -> Linear -> GELU -> Linear` architecture. Its input dimension is matched to the MLLM's hidden state dimension, and its output dimension is aligned with DINOv2's feature dimension (1024). The specific input dimensions are: 4096 for Vicuna-7B and InternVL3.5-8B, 5120 for Vicuna-13B, and 3584 for Qwen2.5-VL-7B. All linear layers were initialized using the Kaiming He initialization scheme (He et al., 2015).

- **LoRA Configuration:** For efficient fine-tuning of the LLM, we applied LoRA (Hu et al., 2022). The configuration was set to a rank ($r$) of 64, `lora_alpha` of 128, and a `lora_dropout` of 0.05. We targeted all linear attention projection layers (`q_proj`, `k_proj`, `v_proj`, `o_proj`) within the language model, as they are crucial for adapting the model's reasoning and attention patterns.

## A.3 TRAINING HYPERPARAMETERS

For all instruction fine-tuning experiments, we maintained a consistent set of training hyperparameters for both the baseline and VISA-enhanced models to ensure a fair and direct comparison. The detailed settings are provided in Table 3.

Table 3: Detailed hyperparameters used during the instruction fine-tuning stage for all models.

| Hyperparameter | Value |
|---|---|
| *General Training Settings* | |
| Total Training Epochs | 1 |
| Global Batch Size | 128 |
| Floating Point Precision | BF16 |
| Gradient Clipping Norm | 1.0 |
| *Optimization Settings* | |
| Optimizer | AdamW (Loshchilov & Hutter, 2017) |
| AdamW Betas $(\beta_1, \beta_2)$ | (0.9, 0.999) |
| AdamW Epsilon $(\epsilon)$ | $1 \times 10^{-8}$ |
| Base Learning Rate | $2 \times 10^{-5}$ |
| Learning Rate Scheduler | Cosine Decay |
| Warmup Ratio | 0.03 (of total steps) |
| Weight Decay | 0.1 |
| *VISA Specific Settings* | |
| Anchoring Strength $(\lambda)$ | 1.0 |
| Structural Loss Weight $(\alpha)$ | 0.5 |

## B DATASET AND EVALUATION DETAILS

This section provides a comprehensive overview of the datasets used for our experiments and the specific protocols followed for evaluation, ensuring transparency and enabling accurate comparison with prior and future work.

### B.1 DATASET DETAILS

We selected a diverse suite of six benchmarks to rigorously assess the fine-grained perceptual and reasoning capabilities of the MLLMs. All images were pre-processed by resizing them while maintaining their original aspect ratio to fit the input resolution requirements of each model's vision encoder.

- **GQA** (Hudson & Manning, 2019): A benchmark designed to evaluate visual reasoning and compositional question answering. It requires understanding spatial relationships, object attributes, and their interactions. We report results on the balanced `testdev` split, which contains 12,578 question-image pairs.

- **MMVP** (Tong et al., 2024c): A targeted benchmark designed to probe the fine-grained visual perception capabilities of MLLMs. It consists of meticulously crafted yes/no questions that challenge a model's ability to discern subtle visual details that are often missed. We evaluate on the full test set of 200 question-image pairs.

- **TextVQA** (Singh et al., 2019): An OCR-VQA benchmark that assesses a model's ability to read and reason about text present in images. This requires a tight integration of visual text recognition and language understanding. We report performance on the official validation set, which includes 5,000 questions.

- **POPE** (Li et al., 2023): A benchmark specifically designed to measure object hallucination. It formulates evaluation as a binary (yes/no) classification task, querying the model about the existence of objects in an image. We evaluate on all three splits (random, popular, adversarial) of the COCO validation set, each containing 3,000 samples, and report the average performance.
- **MME** (Yin et al., 2023): A comprehensive benchmark for evaluating the perception and cognition capabilities of MLLMs across 14 diverse sub-tasks. These tasks span areas such as object existence, counting, position, color, OCR, and more. We evaluate on the full benchmark, which contains approximately 2,800 question-image pairs.
- **MM-Star** (Chen et al., 2024a): A challenging, broad-spectrum benchmark designed to test fine-grained visual understanding across 18 difficult tasks, including attribute recognition, counting, spatial relations, and logic puzzles. We evaluate on the official validation set, comprising 1,800 question-image pairs.

## B.2 Evaluation Protocols

For all benchmarks, we strictly adhered to their official evaluation protocols and utilized their provided evaluation scripts to ensure consistency and comparability with other published results.

- For **GQA**, we report the overall **Accuracy** metric, calculated by the official evaluation server.
- For **MMVP**, which uses a yes/no question format, we report the **Accuracy** of the binary responses.
- For **TextVQA**, we report the standard **VQA Accuracy**, which measures the exact match between the model's generated answer and the ground-truth answer.
- For **POPE**, we report the average **Accuracy** and **F1-Score** across the random, popular, and adversarial splits, as is standard for this benchmark.
- For **MME**, we report the **Total Score**, which is the sum of scores from the Perception and Cognition categories. Each of the 14 sub-tasks is scored on a scale of 0-100, resulting in a maximum possible score of 2800.
- For **MM-Star**, we report the overall **Average Accuracy** across all 18 sub-tasks, as defined by the benchmark's protocol.

## C   Deeper Dive into VISA's Components

This section provides a more detailed examination of the key components of the VISA framework. We elaborate on the specific architecture of the semantic adapter, offer an expanded analysis of our choice of a vision-native anchor, and delve into the formulation of our composite loss, including a broader analysis of hyperparameter sensitivity.

## C.1   Semantic Adapter Architecture

The semantic adapter ($P_\pi$) is a critical component that bridges the MLLM's internal representation space with the VFM's anchor space. It is intentionally designed to be lightweight to avoid introducing significant parameter overhead and to prevent overfitting during instruction tuning.

The adapter is a 3-layer Multi-Layer Perceptron (MLP) with a `Linear -> GELU -> Linear -> GELU -> Linear` architecture. All linear layers are initialized using the Kaiming He initialization scheme (He et al., 2015). The input and output dimensions are dynamically matched to the specific MLLM backbone and the VFM anchor. This design ensures that the adapter is expressive enough to learn the complex transformation between feature spaces while remaining computationally efficient.

## C.2   Anchor VFM Selection: The Case for Vision-Native Representations

The choice of the semantic anchor is the most critical design decision in the VISA framework. Our central hypothesis is that the ideal anchor must provide a high-quality, structured, and *vision-native* feature space. To validate this, we conducted an extensive ablation study, the results of which are presented in Table 2.

- **Language-Aligned Anchor (CLIP-Text):** By forcing the MLLM's intermediate visual representations to align with text embeddings, we observe a significant degradation in performance across all benchmarks. This is because text embeddings, while semantically rich, lack the fine-grained spatial and structural information inherent to visual data. This alignment objective effectively exacerbates semantic attenuation by encouraging the model to discard visual fidelity in favor of linguistic abstraction.

- **Self-Anchor (CLIP-Vision):** This objective encourages consistency between the initial visual projection and the intermediate representations, preventing excessive representational drift. However, its effectiveness is fundamentally capped by the representational limitations of the CLIP encoder itself, which, being trained on image-text pairs, already exhibits a degree of linguistic bias.

- **Vision-Native Anchor (DINOv2):** The DINOv2 model, trained exclusively on images via self-supervision, provides a feature space optimized purely for visual semantics. As shown in the table, aligning with this vision-native anchor yields the most significant and consistent performance improvements. Its representations are rich in the very structural, textural, and spatial details that are often lost during text-centric training, making it an ideal source of grounding for fine-grained visual reasoning.

### C.3 COMPOSITE LOSS FORMULATION AND HYPERPARAMETER SENSITIVITY

**Saliency Weighting from [CLS] Token.** A key component of our point-wise semantic loss ($\mathcal{L}_{\text{point}}$) is the saliency weighting mechanism. In Vision Transformers like DINOv2, the special [CLS] token is designed to aggregate global information from all patch tokens throughout the network. Consequently, the attention scores from the [CLS] token to the various patch tokens in the final self-attention layer provide a natural and powerful proxy for visual saliency. Regions of the image that are critical to its overall semantic meaning receive higher attention from the [CLS] token. By normalizing these attention scores with a Softmax function, we create a weight vector $w$ that allows our loss function to prioritize alignment in the most semantically meaningful regions of the image, while down-weighting less important areas like uniform backgrounds. A visualization of these attention maps will be provided in a later section.

**Hyperparameter Sensitivity Analysis.** As shown in the main paper, VISA is robust to the choice of its main hyperparameters, $\lambda$ and $\alpha$. To further demonstrate this robustness, we extend this sensitivity analysis to our strong backbone, InternVL3.5-8B. Figure 7 show the performance on both the GQA and MMVP benchmarks while varying $\lambda$ (left panels) and $\alpha$ (center panels), alongside an analysis of training convergence (right panels). Similar to our findings with the Vicuna-based model, InternVL exhibits stable performance across a wide range of hyperparameter values, with performance peaking around our selected defaults of $\lambda = 1.0$ and $\alpha = 0.5$. This confirms that VISA is not reliant on brittle hyperparameter tuning and can be effectively applied to different architectures with minimal overhead. The rightmost panels in both figures also clearly illustrate VISA's impact on training efficiency, showing that the VISA-enhanced models consistently outperform their baselines at every stage of training.

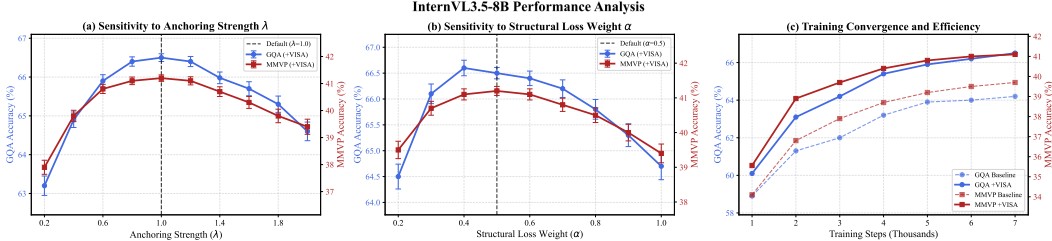

Figure 7: Hyperparameter sensitivity and training efficiency analysis for **InternVL3.5-8B**. Panels (a) and (b) show performance on GQA and MMVP benchmarks as a function of anchoring strength ($\lambda$) and structural loss weight ($\alpha$), respectively. Panel (c) compares the training convergence of the baseline model against the VISA-enhanced version. Dashed lines in (a) and (b) indicate the default values used in our main experiments ($\lambda = 1.0, \alpha = 0.5$). The plots confirm the robustness and efficiency benefits of the VISA framework on a more powerful backbone model.

## C.4 DETAILED FORMULATION OF COMPOSITE LOSS

To provide a clearer understanding of our composite loss, this section offers a more detailed breakdown of its two key components: the saliency-weighted point-wise semantic loss and the structural consistency loss.

**Extracting Saliency Weights from DINOv2.** Our point-wise loss, $\mathcal{L}_{\text{point}}$, is weighted to focus on the most semantically salient regions of an image. We derive these weights directly from the attention mechanism of the DINOv2 anchor model, leveraging its inherent ability to identify foreground objects. Specifically, we consider the multi-head self-attention (MHSA) module in the final transformer block of DINOv2.

Let $H$ be the number of attention heads. For each head $h \in \{1, ..., H\}$, the attention scores between the query of the `[CLS]` token ($Q_{cls}^{(h)}$) and the keys of all $N$ patch tokens ($K_{patches}^{(h)}$) are computed. We average these scores across all heads to obtain a single, robust saliency score for each patch token:

$$A_{\text{avg},i} = \frac{1}{H} \sum_{h=1}^{H} \frac{(Q_{cls}^{(h)})(K_{patches,i}^{(h)})^T}{\sqrt{d_k}} \tag{6}$$

where $A_{\text{avg},i}$ is the averaged attention score for the $i$-th patch token, and $d_k$ is the dimension of the keys per head. This vector of average attention scores, $A_{\text{avg}} \in \mathbb{R}^N$, represents the raw saliency map. Finally, we apply a Softmax function to normalize these scores into a probability distribution, which serves as our weight vector $w$:

$$w_i = \frac{\exp(A_{\text{avg},i})}{\sum_{j=1}^{N} \exp(A_{\text{avg},j})} \tag{7}$$

This process is summarized in Algorithm 1. This method allows us to generate a high-quality saliency map without requiring any external models or labels, relying solely on the emergent properties of the self-supervised VFM.

---

**Algorithm 1** Saliency Weight Extraction from VFM

---

**Require:** VFM final layer features $F_{VFM} \in \mathbb{R}^{(N+1) \times d}$, Number of heads $H$
1: $Q_{cls} \leftarrow \text{ExtractCLSQuery}(F_{VFM})$               ▷ Shape: $(H, d_k)$
2: $K_{patches} \leftarrow \text{ExtractPatchKeys}(F_{VFM})$            ▷ Shape: $(H, N, d_k)$
3: $A_{raw} \leftarrow \text{BatchMatMul}(Q_{cls}, K_{patches}^T)/\sqrt{d_k}$        ▷ Shape: $(H, N)$
4: $A_{avg} \leftarrow \text{Mean}(A_{raw}, \dim = 0)$     ▷ Average scores across heads. Shape: $(N, )$
5: $w \leftarrow \text{Softmax}(A_{avg})$         ▷ Normalize to get final weights. Shape: $(N, )$
6: **return** $w$

---

**Structural Consistency via Gram Matrix and Frobenius Norm.** The structural consistency loss, $\mathcal{L}_{\text{struct}}$, ensures that the relational geometry of the MLLM's feature space mirrors that of the VFM's. This is achieved by comparing their respective Gram matrices.

The computation of the Gram matrix for a set of feature vectors (e.g., the VFM's target features $Y \in \mathbb{R}^{N \times d}$) involves two steps:

1. **L2 Normalization:** Each feature vector $y_i$ in $Y$ is normalized to have a unit L2 norm: $\hat{y}_i = y_i / \|y_i\|_2$. This step is crucial as it isolates the orientation of the vectors from their magnitude. The resulting matrix of normalized vectors is $\hat{Y}$.

2. **Self-Similarity Computation:** The Gram matrix is computed as the matrix product of the normalized feature set with its transpose: $G(Y) = \hat{Y}\hat{Y}^T$.

The resulting Gram matrix $G(Y) \in \mathbb{R}^{N \times N}$ is a symmetric matrix where each element $G(Y)_{ij}$ represents the cosine similarity between the feature vectors of the $i$-th and $j$-th tokens. It serves as a comprehensive fingerprint of the entire relational structure of the feature space.

We use the squared Frobenius norm to measure the distance between the Gram matrix of the VFM ($G(Y)$) and that of the MLLM's adapted features ($G(E'_\ell)$). The Frobenius norm is the matrix

equivalent of the vector L2 norm and is defined as $\|A\|_F = \sqrt{\sum_{i,j} a_{ij}^2}$. We choose it for several reasons:

- **Differentiability:** It is a smooth and differentiable function, making it well-suited for gradient-based optimization.
- **Holistic Comparison:** Unlike other matrix norms that might only consider singular values, the Frobenius norm accounts for the difference between every single corresponding element in the two matrices. This ensures that the loss penalizes any deviation in the pairwise similarity structure, forcing a holistic alignment of the feature space geometries.
- **Interpretability:** The squared Frobenius norm loss is equivalent to the mean squared error (MSE) over all pairwise cosine similarities, providing a clear and interpretable objective: to make the relational structure of the MLLM's visual features as close as possible to that of the VFM anchor.

**Synergy of Composite Loss.** The two loss components are designed to be complementary. $\mathcal{L}_{\text{point}}$ acts as a semantic guide, pulling the MLLM's representations of salient objects towards their correct locations in the anchor's feature space. Concurrently, $\mathcal{L}_{\text{struct}}$ acts as a structural regularizer, ensuring that the relationships *between* all tokens (e.g., object-to-object and object-to-background similarities) are preserved. Together, they ensure a comprehensive alignment that captures both fine-grained object details and the broader compositional structure of the scene.

## D   DIAGNOSTIC PROBE PROTOCOLS (SUPPLEMENT TO SECTION 3.2)

To empirically validate the phenomenon of Semantic Attenuation, we conducted layer-wise probing experiments. Here we detail the dataset construction, probe architecture, and training protocol used to generate the results in Figure 2.

**Dataset Construction.** We constructed a diagnostic dataset derived from the GQA dataset (Hudson & Manning, 2019), chosen for its rich semantic annotations regarding object attributes and spatial relationships. We curated two specific subsets to test fine-grained visual perception:

- **Object Counting:** We balanced the dataset to ensure a uniform distribution across counts to prevent the probe from learning class priors.
- **Spatial Relation:** We selected questions involving spatial prepositions. We formulated this as a classification task where the probe must predict the spatial relationship given the global image feature.

**Probe Architecture and Training.** For a specific layer $l$ in the MLLM's visual encoder (or LLM layers), we extracted the feature representations $E_l$. The probe consists of a lightweight 2-layer MLP (Linear $\rightarrow$ ReLU $\rightarrow$ Linear) mapping the frozen feature dimension to the task label space.

- **Optimization:** We trained each probe for 10 epochs using the AdamW optimizer with a learning rate of $1e^{-3}$ and a batch size of 256.
- **Metric:** We report top-1 accuracy on the validation set.

The results (Figure 2) consistently show that while early layers maintain high probing accuracy for counting and spatial tasks, performance degrades significantly in deeper layers of the baseline model, confirming the loss of fine-grained visual information.

## E   FEATURE FREQUENCY EVOLUTION ANALYSIS

To empirically verify the hypothesis that text-only supervision leads to the progressive loss of fine-grained visual details, we conducted a motivational experiment tracking the evolution of feature frequency characteristics throughout the instruction tuning process.

### E.1   DEFINITION OF HIGH-FREQUENCY INFORMATION

Following previous studies on the spectral analysis of Vision Transformers (Park et al., 2023), we quantify the amount of detail in a feature map using the energy distribution in the frequency domain.

Let $F \in \mathbb{R}^{H \times W}$ be a single channel of a 2D feature map from the visual encoder or an intermediate layer of the LLM. We compute its 2D Discrete Fourier Transform (DFT), denoted as $\mathcal{F}(u, v)$, where $(u, v)$ represent the spatial frequencies. We shift the zero-frequency component to the center of the spectrum. The High-Frequency Ratio (HFR) is defined as the proportion of the spectral energy residing in high-frequency regions relative to the total energy:

$$\text{HFR} = \frac{\sum_{u,v \in \Omega_{\text{high}}} |\mathcal{F}(u, v)|^2}{\sum_{u,v \in \Omega_{\text{all}}} |\mathcal{F}(u, v)|^2} \tag{8}$$

where $\Omega_{\text{all}}$ is the set of all frequency coordinates, and $\Omega_{\text{high}}$ represents the high-frequency region. Specifically, we define $\Omega_{\text{high}}$ as the region where the radial frequency distance from the center $\sqrt{u^2 + v^2}$ is greater than a threshold $\tau$. In our experiments, we set $\tau = \frac{2}{3} \times \max(\sqrt{u^2 + v^2})$, focusing on the top 33% highest frequencies which typically encode edges, textures, and fine-grained details. The final HFR for a layer is averaged across all feature channels and all samples in the evaluation batch.

### E.2 EXPERIMENTAL SETUP

To monitor the evolution of HFR during training, we adopted the following protocol:

- **Models:** We compared two models: (1) The Baseline (LLaVA-1.5-7B) trained with standard text-only supervision, and (2) Our VISA-enhanced model trained with the additional visual semantic anchoring loss. Both models were initialized from the same pre-trained weights.

- **Tracking Interval:** We performed Supervised Fine-Tuning (SFT) for 1 epoch on the LLaVA-665K dataset. We saved checkpoints every 500 training steps.

- **Evaluation Probe:** At each checkpoint, we extracted visual representations from the 16th layer of the LLM (the target layer for our anchoring) using a held-out subset of 500 images from the GQA dataset.

- **Metric Calculation:** We computed the HFR for these extracted features and averaged them to obtain a single value for that training step.

## F   FINE-GRAINED PERCEPTION ANALYSIS ON GVTBENCH

To further address the need for explicit fine-grained perception evaluation raised by Reviewer BibD, we conducted comprehensive experiments using the **GVTBench** framework (Wang et al., 2023). Unlike general VQA benchmarks, GVTBench is specifically curated to disentangle visual semantic understanding from fine-grained perception capabilities using a standardized codebase and evaluation protocol.

**Task Definitions.** We utilize the two core tasks supported by the GVTBench codebase that are most indicative of fine-grained visual grounding:

- **Object Counting (OC):** The model must accurately count specific objects in the image. This task demands precise instance-level discrimination and is highly sensitive to the semantic attenuation of high-frequency details.

- **Multi-Class Identification (MCI):** This task tests the model's ability to verify the simultaneous existence of multiple specific objects, serving as a rigorous test for multi-object recognition without spatial cues.

**Results.** The results are summarized in Table 4. We observe that **VISA** yields significant performance gains, particularly on the Object Counting task which is notoriously difficult for standard CLIP-based MLLMs. For the LLaVA-1.5-7B backbone, VISA improves Object Counting accuracy by **+4.4%** (38.4% → 42.8%), effectively mitigating the counting deficits caused by the loss of fine-grained visual tokens. For MCI, where the baseline is already strong due to CLIP's robust semantic priors, VISA still achieves a consistent improvement of **+2.2%**.

Table 4: **Fine-grained perception performance on GVTBench.** We report accuracy (%) on Object Counting (OC) and Multi-Class Identification (MCI) tasks averaged across COCO and VCR splits. The results demonstrate that VISA consistently enhances fine-grained perception on the LLaVA-1.5 architecture.

| Base Model | Method | GVTBench Tasks | | Avg. |
| | | Object Counting | MCI | |
|---|---|---|---|---|
| LLaVA-1.5-7B | Baseline | 38.4 | 81.5 | 60.0 |
| | **+ VISA (Ours)** | **42.8** (+4.4) | **83.7** (+2.2) | **63.3** |
| LLaVA-1.5-13B | Baseline | 42.1 | 83.2 | 62.7 |
| | **+ VISA (Ours)** | **45.3** (+3.2) | **85.1** (+1.9) | **65.7** |

## G  VISA AS AN EFFICIENT IMPLICIT ENSEMBLE

A key insight of our work is that VISA acts as a form of Implicit Ensemble. Recent works like BRAVE (Kar et al., 2024) and MoVE-KD (Cao et al., 2025) advocate for Explicit Ensembles, where features from multiple encoders (e.g., CLIP + DINOv2) are concatenated and fed to the LLM. While effective, this approach significantly increases computational costs.

A critical advantage of VISA is its efficiency compared to Explicit Ensemble methods (e.g., BRAVE, MoVE-KD) which require running multiple vision encoders during inference. We provide a detailed breakdown of computational costs in Table 5.

To quantify the efficiency gains of VISA, we implemented an Explicit Ensemble baseline that concatenates CLIP and DINOv2 features (doubling the visual token count) and trained it under the same settings. Table 5 presents the comparison.

Table 5: We compare our Implicit Ensemble method (VISA) against the Explicit Ensemble (CLIP+DINOv2). In training and inference, VISA maintains the efficiency while recovering ~95% of the Explicit Ensemble's fine-grained perception gains.

| Method | Training Phase | Inference Phase | | Performance ↑ | |
| | Memory | FLOPs | Latency | GQA | MMVP |
|---|---|---|---|---|---|
| **Baseline** | 42.5 GB | 1.00× | 28ms | **61.2** | **28.0** |
| **Explicit Ensemble** (CLIP + DINOv2) | 58.4 GB | 1.42× | 45ms | **65.1** | **34.3** |
| **VISA (Ours)** | **47.2 GB** | **1.00×** | **28ms** | 64.2 | 32.9 |

## H  QUALITATIVE ANALYSIS AND VISUALIZATIONS

To provide a more intuitive understanding of how VISA enhances the internal mechanisms of MLLMs, this section presents a series of qualitative analyses. We focus on visualizing the model's internal attention distributions and the frequency characteristics of its learned visual representations to demonstrate the tangible impact of our framework on fine-grained perception.

### H.1  ATTENTION MAP AND FOURIER SPECTRUM ANALYSIS

We investigate the internal representations at Layer 16 of our models to visualize the effects of VISA. We employ two analytical techniques:

- **Attention Maps:** These maps are generated from the model's final attention layer and are overlaid on the input image. They reveal which regions of the image the model focuses on when generating its response. A more detailed and accurately focused attention map is indicative of superior fine-grained perception.

- **Fourier Spectra:** We compute the 2D Fourier transform of the visual feature maps. The resulting spectrum visualizes the frequency components of the representation. A spectrum heavily concentrated at the center signifies a bias towards low-frequency information (coarse details), a hallmark of semantic attenuation. Conversely, a broader spectrum with more energy in the outer regions indicates the preservation of high-frequency details (fine textures, edges, and small objects).

The following examples in Figures 8, 9, and 10 compare the baseline model with the VISA-enhanced version on challenging visual reasoning tasks. In each case, VISA's ability to preserve high-frequency information, as evidenced by its Fourier spectrum, directly translates to a more precise and detailed attention map, enabling it to correctly answer questions where the baseline model fails.

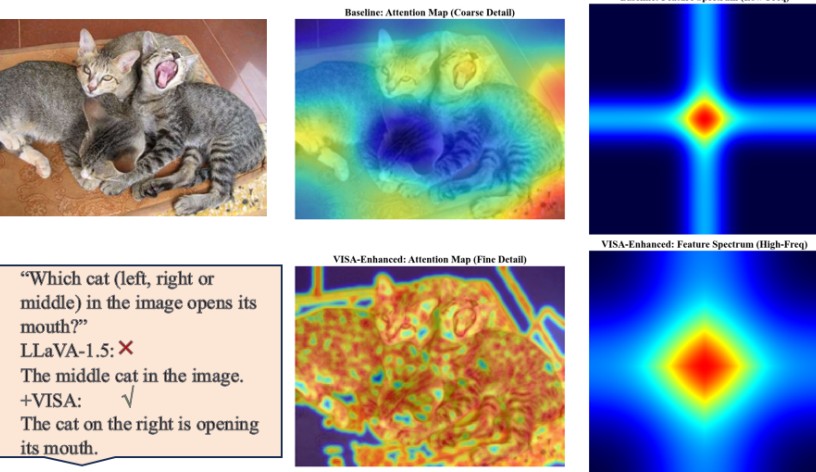

Figure 8: Analysis on a fine-grained state recognition task. The question, Which cat (left, right or middle) in the image opens its mouth?, requires the model to distinguish between similar objects and identify a specific state. The baseline model's diffuse attention map and low-frequency spectrum indicate a loss of detail, leading to an incorrect answer. In contrast, VISA's attention is sharply focused on the correct cat's head.

In the scenario presented in Figure 8, the task demands not only spatial reasoning but also the recognition of a subtle action. The baseline model, suffering from semantic attenuation, generates a coarse, generalized attention map that covers both cats without distinguishing between them. Its reasoning is therefore not grounded in the specific visual evidence, resulting in a factual error. VISA, by anchoring the representation to a high-fidelity visual space, preserves the necessary details. Its attention map is precise, and its Fourier spectrum confirms the presence of high-frequency features, allowing it to correctly identify both the location and the state of the target object.

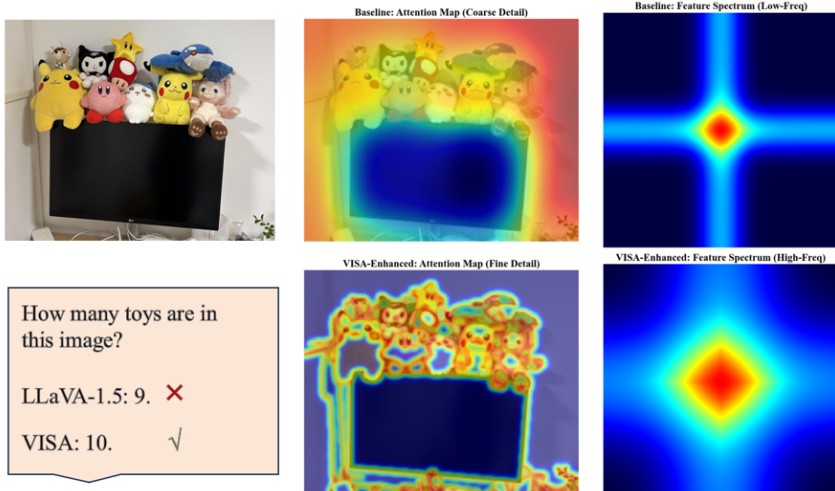

Figure 9: Analysis on a challenging object counting task. The question, How many toys are in this image?, requires the model to segment and count multiple small, cluttered objects. The baseline model's attention is blurry and merges the dolls, a classic symptom of semantic attenuation, leading to an incorrect count. VISA's attention map shows distinct, sharp activations for each individual toy.

Figure 9 illustrates a common failure mode for MLLMs: counting densely packed objects. The baseline model's internal representation, biased towards low frequencies, is unable to maintain the distinct boundaries between the toys. Its attention map treats the group of dolls as a single entity, making an accurate count impossible. The VISA-enhanced model, however, successfully preserves the high-frequency edge information. This allows its attention mechanism to individually segment each toy, leading to a correct count of ten. This example underscores how preserving fundamental visual details is a prerequisite for more complex cognitive tasks like enumeration.

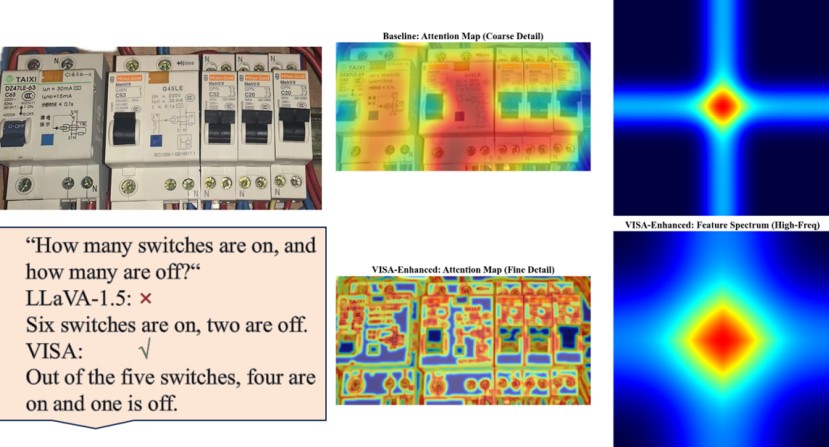

Figure 10: Analysis on a complex, multi-part reasoning task involving both counting and state classification. The question, How many switches are on, and how many are off?, requires extremely fine-grained perception. The baseline's attention is coarse and fails to capture the subtle state of each switch. VISA's attention map is remarkably detailed, highlighting the precise orientation of each individual lever.

The task in Figure 10 is the most demanding, as it requires not only counting but also classifying the state (on/off) of multiple small components based on their orientation. The baseline model's attention is spread generally across the panel of switches, failing completely to resolve the state of any individual switch. This demonstrates a severe case of semantic attenuation, where all fine-grained details have been lost. VISA, in stark contrast, produces a highly detailed attention map that precisely localizes each switch's lever. This demonstrates that by preventing the loss of high-frequency visual information, VISA enables the MLLM to perform complex, real-world perceptual tasks that are far beyond the capabilities of models trained with indirect, text-only supervision.

