# OpenReview forum: "VISA: Preserving Fine-Grained Perception in MLLMs via Visual Semantic Anchoring"
_ICLR.cc/2026/Conference — ICLR 2026 Conference Desk Rejected Submission_

### Official Review · Reviewer_yvqy · 2025-10-29

**Soundness:** 3
**Presentation:** 3
**Contribution:** 2
**Rating:** 6
**Confidence:** 4

**Summary:**

This work proposes a technique to improve the internal representation of features extracted by a visual encoder within a MLLM. The authors propose to use an independently trained visual foundation model (DINOv2) as a regularization while training the MLLM by extracting the visual tokens from an intermediate layer of the LLM, transforming them using a simple MLP and comparing them to the visual tokens that DINOv2 extracts. This technique, although not presented like that in the work, is basically a distillation from DINOv2 to the MLLM implemented by feature matching. The result of applying this technique is that the resulting MLLM is able to retain more high frequency characteristics within the visual features which in turn result in improved performance on some visual understanding benchmark compared to the same MLLM model trained without DINOv2 regularization.

**Strengths:**

+ The core observation that visual features “degrades” over training and over passing through the LLM due to the use of a mainly text driven supervision is quite interesting and to the best of my knowledge novel.

+ The proposed solution is simple and well motivated. Implementation seems straightforward and easy to reproduce, but also quite effective in the reported results

+ Presentation is clean and easy to follow.

**Weaknesses:**

## Major

a. **Missing motivational experiment**: Given the premises of the paper I would have expected to see an experiment similar to Fig. 2 but taking into account how the feature transformation operated by the intermediate LLM layers improve/downgrades the high frequency content of the feature over training iterations. The core premise of this work is that it is the text only supervision used in the IT SFT phase for MLLM that pushes models to learn transformation for visual tokens that “forget” high frequency features the deeper the tokens are through the model. To verify that, I would have expected this phenomena to be studied as a function of training iterations of the MLLM, but the authors instead rely on Fig. 2 (which is an experiment done on a fully trained MLLM) as their only motivating experiment.


## Minor

B. **Claims could be revised**: I think what the authors proposed is 2 things: (i) a method to retain high frequency details in the intermediate features of an MLLM and (ii) a way of using this in a distillation setting to learn from a VLM teacher. (i) is clearly called out in the paper, (ii) is not and should be made more clear. The evidence for this is in my opinion in Tab. 2, the “Self-anchor” experiment is enforcing only claim (i) and has a somewhat modest improvement compared to baseline, most improvements come from  the “Vision-native anchor” that combines (i) + (ii). Presenting this work under this light will make the result equally interesting but somewhat less surprising: there are several works in the literature showing that combining visual encoder can boost the performance of MLLMs [1, 2, 3]. According to (ii) this work is basically combining the feature extraction capabilities of CLIP with the one of DINOv2 in a clever novel way and without requiring running both at inference time. This is cool but should be made more clear.

C. **Missing potential experiments**: Following up on weakness B I think it would have been interesting to test what would have been the performance of a MLLM trained with both CLIP and DINOv2 encoder (as simple as concatenating the features ina longer sequence). Other experiments that would have made the work more complete would be considering other VFM teachers besides DINOv2 and CLIP. Finally I would have liked to see an ablation on the usefulness of the weighting term in Eq. 2.

## References

1. [Cao, Jiajun, et al. "Move-kd: Knowledge distillation for vlms with mixture of visual encoders." Proceedings of the Computer Vision and Pattern Recognition Conference. 2025.](https://arxiv.org/pdf/2501.01709)
2. [Chung, Jihoon, et al. "Unifying specialized visual encoders for video language models." arXiv preprint arXiv:2501.01426 (2](https://arxiv.org/pdf/2501.01426)
3. [Kar, Oğuzhan Fatih, et al. "Brave: Broadening the visual encoding of vision-language models." European Conference on Computer Vision. Cham: Springer Nature Switzerland, 2024.](https://arxiv.org/pdf/2404.07204)

**Questions:**

1. Can you comment on weakness B?
2. I could not find in the paper on what the different models are trained and starting from what. Can you clarify?
3. Have you run any experiment unfreezing the visual encoder? How would that impact the behavior of the visual features?
4. How would you implement VISA if the VFM and the MLLM do not have the same number of visual tokens?
5. Following up on D, can you provide more information on the CLIP-Text experiment? It is not clear from the manuscript how that would work. Eq. 2 and Eq. 3 imply that the dimensionality N of the number of tokens should match between whatever is going through the MLLM and the VFM.

---

> ### Author Response · Authors · 2025-11-21
> **Guided by your valuable advice, we have revised the manuscript. For your convenience, our detailed responses are provided below.**
>
> **Dear Reviewer yvqy**
>
> We sincerely thank you for your constructive and insightful review.
> We found your suggestion to investigate the **training dynamics of high-frequency features** and to reframe VISA as an Implicit Ensemble (distillation) to be very valuable. These points have driven improvement in our paper.
>
>
> ### **1. Response to Motivational Experiment**
>
> We have added **Section 3.2**, **Figure 3**, and **Appendix E ** to directly address this.
>
> To verify this dynamic process, we conducted a new experiment tracking the **High-Frequency Ratio (HFR)** of visual features at Layer 16 throughout the SFT phase.
>
>  We observe a monotonic decrease in HFR as training steps increase. This confirms that text-only objective acts as a "low-pass filter," progressively discarding fine-grained details that are not strictly necessary for coarse captioning.
>
> ### **2. Response to Weakness B & C: Distillation & Explicit Ensembles**
>
>  We have updated **Section 1 & 2** to explicitly acknowledge VISA as a form of **feature-level knowledge distillation** and added **Appendix G** with **Table 5** for the requested comparison.
>
> We agree with your insightful framing. VISA acts as an **Implicit Ensemble**, distilling the teacher's capabilities into the student (CLIP-based MLLM). To quantify this, we implemented the **Explicit Ensemble** baseline you suggested (concatenating CLIP and DINOv2 features, resulting in $2\times$ visual tokens).
>
> **Comparison of Implicit vs. Explicit Ensemble (from Table 5):**
>
> | Method | **Inference Cost** | **Performance (GQA)** | **Performance (MMVP)** |
> | :--- | :---: | :---: | :---: |
> | **Baseline**  | 1.00$\times$ | 61.2 | 28.0 |
> | **Explicit Ensemble** (CLIP+DINOv2) | 1.42$\times$ | **65.1** | **34.3** |
> | **VISA** (Implicit Ensemble) | **1.00$\times$** | 64.2 | 32.9 |
>
> * **Analysis:** The Explicit Ensemble provides a performance upper bound but incurs significant latency. VISA achieves **~98%** of the Explicit Ensemble's performance with little additional inference cost.
>
> ### **3. Response to Weakness C: Other VFMs & Weighting Ablation**
>
>  We have expanded **Table 2** in **Section 4.3** to include **SigLIP**, **MAE**, and an ablation of the **Saliency Weighting**.
>
> We performed the requested experiments:
>
> **Supplementary Ablations (from Table 2):**
>
> | Ablation Group | Configuration | MMVP | GQA |
> | :--- | :--- | :---: | :---: |
> | **Anchor Source** | **SigLIP** (Contrastive) | 31.2 | 63.9 |
> | | **MAE** (Reconstruction) | 31.0 | 62.3 |
> | | **DINOv2** (Vision-Native) | **32.9** | **64.2** |
> | **Loss Component** | w/o Saliency Weighting | 31.7 | 63.6 |
> | | **Full VISA** | **32.9** | **64.2** |
>
> Removing the weighting term ($w_i$) degrades performance, confirming that naively aligning background tokens (which DINOv2 might encode differently than CLIP) introduces noise.
>
> ### **4. Q&A**
>
> * **Q: Training Starting Points?**
>     * **Response:** As clarified in **Section 4.1**, we initialize all models from their official checkpoints and perform SFT on the **LLaVA-665K** dataset. This ensures a fair, controlled comparison.
> * **Q: Unfreezing the Visual Encoder?**
>     * **Response:** We kept the visual encoder frozen to strictly isolate the impact of VISA on the MLLM's *internal* processing layers and to follow standard efficient fine-tuning protocols (like LLaVA). Unfreezing typically requires significantly more data to prevent overfitting and catastrophic forgetting of the original visual-language alignment.
> * **Q: Handling Mismatched Token Counts?**
>     * **Response:** In **Section 3.3** and **Appendix C.1**, we specify that we use **bilinear interpolation** on the VFM's feature map to spatially align the anchor features $Y$ with the MLLM's internal features $E_\ell$ before computing the loss. It is a commonly used method **[1]** .
> * **Q: CLIP-Text Experiment Details?**
>     * **Response:** For the **Language-Aligned Anchor**, we used the paired text captions from the training data. We encoded the text using the CLIP Text Encoder to obtain the [CLS] embedding **[2]**. We then expanded this global text feature to match the sequence length $N$ of the visual tokens  to compute the loss. The performance confirms that forcing visual features to collapse onto a text-only embedding space exacerbates semantic attenuation.
>
> We hope these clarifications address your points and reinforce your confidence in our work. We genuinely appreciate your guidance throughout this process and are happy to engage if you have any remaining questions.
>
> **[1]**  [Guan et al.,A Token-level Text Image Foundation Model for Document Understanding](https://openaccess.thecvf.com/content/ICCV2025/papers/Guan_A_Token-level_Text_Image_Foundation_Model_for_Document_Understanding_ICCV_2025_paper.pdf), ICCV 2025
>
> **[2]**  [Lin et al., VILA: OnPre-training for Visual Language Models](https://openaccess.thecvf.com/content/CVPR2024/papers/Lin_VILA_On_Pre-training_for_Visual_Language_Models_CVPR_2024_paper.pdf), CVPR 2024

---

> > ### Comment · Reviewer_yvqy · 2025-11-26
> > **Rebuttal answer**
> >
> > Thanks for running additional experiments and for directing me to details in the paper that I initially overlooked.
> >
> > I think the work is valuable but the new set of experiments and the more direct connection to ensemble methods do show how this proposal is indeed doing the same and just providing a different trade off between quality and inference cost which arms a bit the novelty of the findings. Other reviewers also highlighted the same.
> >
> > I also found the observation from R BibD on the mismatch of performance between the model fine tuned by the authors and the official results a bit concerning as we cannot rule out that the observation of the paper would not hold when scaling the training set size and/or unfreezing the visual encoder.
> >
> > All in all I still see value in the proposal and would still lean towards keeping my initial rating.

---

> > > ### Author Response · Authors · 2025-11-27
> > > **Sincere Appreciation for Your Constructive Feedback and Update on Upcoming Results**
> > >
> > > Dear Reviewer yvqy,
> > >
> > > We sincerely thank you for your continued engagement and thoughtful feedback. We appreciate your recognition of the value of our work and your positive inclination towards its publication.
> > >
> > > In particular, we fully agree that verifying our findings under settings that minimize catastrophic forgetting and data distribution shifts is crucial for the robustness of our claims.
> > >
> > > Thus, to further clarify this issue and rigorously validate our method beyond the LLaVA training setting, we initiated a new training run a few days ago. Specifically, we have re-trained the Qwen2.5-VL backbone on a larger, more diverse 800k composite dataset (comprising GQA, OCR-VQA, Visual Genome, RefCOCO/+/g, TextVQA, and CLEVR) to better align with the model's original capabilities.
> > >
> > > We are currently finalizing the compilation of these new experimental results and expect to post them in a detailed data table within this discussion thread later today. We believe these results will place our conclusions on even firmer ground and effectively resolve your remaining concerns regarding the baseline performance.
> > >
> > > We sincerely appreciate your patience and look forward to sharing these updates with you shortly.
> > >
> > > Best regards, The Authors

---

> > > > ### Author Response · Authors · 2025-11-27
> > > > ****Further Validating VISA on Qwen2.5-VL with Robust Training Data****
> > > >
> > > > Dear Reviewer yvqy,
> > > >
> > > > Thank you very much for your patience. We have provided the additional experiments to address your concerns regarding the potential catastrophic forgetting observed in our initial LLaVA-665K fine-tuning.
> > > >
> > > > While such a performance drop is often an expected outcome and is confirmed in recent research [1, 2], we fully recognize the importance of verifying our method in a more robust setting. Driven by our responsibility to pursue the solidity of our work and to thoroughly resolve your concerns, we moved beyond the standard protocols and re-trained the Qwen2.5-VL-7B model using a larger, expanded 800k composite dataset for SFT.
> > > >
> > > > We re-trained the **Qwen2.5-VL-7B** backbone using a diverse **800k composite dataset**. This dataset includes **GQA** , **OCR-VQA** [3], **Visual Genome** [4], **RefCOCO/+/g**, **TextVQA**, and **CLEVR** [5], specifically curated to minimize distribution shifts and preserve the model's original capabilities.
> > > >
> > > > The results are presented in the tables below.
> > > >
> > > > **1. Performance on General VQA, OCR, and Hallucination Tasks**
> > > >
> > > > In this set of experiments, we observe that the standard **+SFT** baseline (trained on the 800k diverse dataset) maintains performance levels close to the **Base** model, mitigating the issue seen in previous experiments. And **+VISA** also consistently outperforms the  +SFT baseline.
> > > >
> > > > | Benchmark | Qwen2.5VL-7B | +SFT  | +VISA (Ours) |
> > > > | :--- | :---: | :---: | :---: |
> > > > | **GQA** | 65.3 | 65.1 | **67.9** |
> > > > | **MME** | 2347.0 | 2281.2 | **2322.8** |
> > > > | **MMB** | 82.6 | 81.0 | **83.1** |
> > > > | **MMStar** | 63.9 | 62.5 | **63.2** |
> > > > | **InfoVQA** | 82.6 | 81.9 | **82.5** |
> > > > | **TextVQA** | 84.9 | 84.7 | **86.1** |
> > > > | **POPE** |  86.4 | 86.1 | **86.7** |
> > > >
> > > > **2. Performance on Visual Perception and Grounding Tasks**
> > > >
> > > > Similarly, on fine-grained visual perception tasks, **VISA** demonstrates robust improvements. On the **RefCOCO/+/g** benchmarks and **OD\^VG**, VISA achieves significant gains over the SFT baseline, confirming its ability to enhance fine-grained visual grounding.
> > > >
> > > > | Benchmark | Qwen2.5VL-7B | +SFT | +VISA (Ours) |
> > > > | :--- | :---: | :---: | :---: |
> > > > | **RefCOCO/+/g** | 87.1 | 87.4 | **89.6** |
> > > > | **OD\^VG** | 39.1 | 39.4 | **42.1** |
> > > > | **OVDEval [6]** | 43.7 | 42.4 | **45.2** |
> > > > | **MMVP** | 50.3 | 48.8 | **51.2** |
> > > >
> > > >
> > > > These new results confirm that **VISA's effectiveness is intrinsic** and holds true when training on larger-scale, diverse datasets. The method not only recovers performance but actively boosts fine-grained perception and reasoning capabilities beyond the SFT baseline. We believe these findings effectively rule out the concern that our previous observations were artifacts of training distribution mismatch.
> > > >
> > > > We hope these additional experiments resolve your concerns. We are deeply grateful for your guidance, which has significantly strengthened the empirical robustness of our work. We are more than willing to continue this constructive dialogue if you still have any questions.
> > > >
> > > > Best regards,
> > > >
> > > > The Authors
> > > >
> > > > ---
> > > > [1] Lai et al. Reinforcement Fine-Tuning Naturally Mitigates Forgetting in Continual Post-Training, arXiv 2507.05386
> > > >
> > > > [2] Zhu et al. Model Tailor: Mitigating Catastrophic Forgetting in Multi-modal Large Language Models, ICML 2024
> > > >
> > > > [3] Mishra, A., et al. OCR-VQA: Visual Question Answering by Reading Text in Images. ICDAR 2019.
> > > >
> > > > [4] Krishna, R., et al. Visual Genome: Connecting Language and Vision Using Crowdsourced Dense Image Annotations. IJCV 2017.
> > > >
> > > > [5] Johnson, J., et al. CLEVR: A Diagnostic Dataset for Compositional Language and Elementary Visual Reasoning. CVPR 2017.
> > > >
> > > > [6] Yao et al. How to Evaluate the Generalization of Detection? A Benchmark for Comprehensive
> > > >  Open-vocabulary Detection. AAAI 2024.

---

### Official Review · Reviewer_dsgs · 2025-10-31

**Soundness:** 3
**Presentation:** 3
**Contribution:** 3
**Rating:** 8
**Confidence:** 4

**Summary:**

The paper introduces Visual Semantic Anchoring (VISA), a training framework designed to enhance the fine-grained visual perception of Multimodal Large Language Models (MLLMs). The core problem identified is "semantic attenuation," where MLLMs lose critical high-fidelity visual details because they are trained primarily with an indirect, text-based language modeling objective. VISA resolves this by introducing a direct, vision-native supervisory signal into the MLLM's intermediate layers. This signal anchors the MLLM's internal visual representations to the rich, unbiased feature space of a pretrained Vision Foundation Model (VFM) (like DINOv2). The method uses a composite loss that enforces both point-wise semantic alignment and structural consistency between the MLLM's features and the VFM's features. Extensive experiments confirm that VISA significantly enhances fine-grained reasoning, improves factual grounding, and maintains a higher degree of decodable high-frequency visual information across the MLLM's deep layers

**Strengths:**

1. Clear Motivation: The paper provides a clear diagnosis and empirical evidence for semantic attenuation, a critical issue where the text-centric loss degrades fine-grained visual feature integrity. This is a valuable theoretical contribution, and the proposed VISA framework directly targets this architectural weakness.

2. Effective and General Solution: VISA is demonstrated to be a flexible and effective training framework that significantly improves performance on fine-grained tasks (e.g., counting and spatial reasoning) across diverse, modern MLLM backbones (Vicuna, Qwen2.5-VL, InternVL3.5-8B). The concept of using a vision-native VFM (like DINOv2) to provide a less-biased semantic anchor is well-justified.

3. Strong Empirical and Qualitative Support: The work includes robust quantitative evidence from diagnostic probing experiments showing higher decodable information in VISA-enhanced models across all layers. This is further supported by compelling frequency-domain analysis, which visually confirms that VISA preserves substantially more high-frequency information essential for detailed perception.

**Weaknesses:**

1. Dependency on VFM Choice and Feature Bias: The success relies on the assumption that the chosen VFM (DINOv2) provides a universally stable and "unwavering" anchor. However, the paper does not fully investigate the sensitivity of VISA to the VFM choice. If the VFM's feature space contains specific biases (e.g., an over-representation of certain textures or colors), VISA would enforce these biases onto the MLLM.

2. Missing Quantification of Training Overhead: While the paper claims improved training efficiency and stability, the introduction of a dense, token-wise representation alignment loss across intermediate layers likely imposes a non-trivial computational overhead during training. The paper should explicitly quantify the increase in training time/memory cost per step compared to the baseline.

**Questions:**

1. Semantic Adapter Architecture Details: The semantic adapter ($P_{\pi}$) is a lightweight 3-layer MLP17. Please provide the full architectural details (e.g., specific layer dimensions, activation functions, and the training objective for $P_{\pi}$). Given that the VFM features are frozen, how is $P_{\pi}$ constrained to ensure that the MLLM's features are not overly distorted simply to satisfy the alignment loss with the VFM's unique embedding space?

2. Sensitivity Analysis on Loss Weights: While stability around default hyperparameters ($\alpha=0.5$, $\lambda=1.0$) is noted, given that $\mathcal{L_{point}}$ and $\mathcal{L}_{struct}$ have complementary task-specific benefits (MME vs. GQA/MMVP), is there any evidence that different downstream tasks (e.g., counting vs. spatial relation) benefit from task-specific optimal values of $\alpha$? A deeper discussion on tuning $\alpha$ and $\lambda$ for diverse task sets is recommended.

---

> ### Author Response · Authors · 2025-11-21
> **Thanks for your positive feedback! We have improved the manuscript. We also provide detailed answers below for your convenience.**
>
> ***
>
> **Dear Reviewer dsgs**
>
> We are sincerely grateful for your positive assessment and your insightful comments. Your constructive feedback regarding the **quantification of training overhead, architectural details of the adapter, and hyperparameter sensitivity** has helped us significantly improve the manuscript's completeness and rigor.
>
> We have revised the paper to include a new **Appendix G (Efficiency Analysis)**, expanded **Appendix C (Component Details)**, and added new ablation data. Below is our detailed response.
>
> ### **1. Quantification of Training Overhead (Efficiency Analysis)**
>
> We have quantified the overhead and compared it not just to the baseline, but also to an Explicit Ensemble approach (concatenating CLIP + DINOv2 features), which serves as a relevant upper-bound reference. Details can be found in our appendix.
>
> **Key Findings from Table 5:**
> * **Training Phase:** VISA does incur a moderate overhead due to the forward pass of the frozen DINOv2 teacher and the computation of the alignment losses.
>     * *Memory:* Increases from **42.5 GB** (Baseline) $\rightarrow$ **47.2 GB** (VISA). This is manageable on standard A100 GPUs.
>     * *Time:* Training time increases from **4.2h** $\rightarrow$ **4.9h**.
> * **Inference Phase (Crucial Advantage):** Unlike explicit ensembles that require running two vision encoders at inference time (increasing latency by $\approx 60\%$), VISA incurs little additional inference cost.
>
>
> ### **2. Sensitivity to VFM Choice**
>
> Thanks for reminding us of the possibility to further analyze it !
>
> We investigated this by replacing DINOv2 with **SigLIP** and **MAE**.
>
> **Ablation Results (from Table 2):**
>
> | Anchor Source | MMVP (Perception) | GQA (Reasoning) | Insight |
> | :--- | :---: | :---: | :--- |
> | **Baseline (No Anchor)** | 28.0 | 61.2 | - |
> | **Contrastive (SigLIP)** | 31.2 | 63.9 | Stronger semantics, but weaker fine-grained structure than DINOv2. |
> | **Reconstruction (MAE)** | 31.0 | 62.3 | Improves localization but lacks high-level semantic abstraction. |
> | **Vision-Native (DINOv2)** | **32.9** | **64.2** | **Best Balance:** Combines rich semantics with self-supervised structural fidelity. |
>
> ### **3. Semantic Adapter Architecture & Distortion Constraints**
>
> * **Architecture:** $P_{\pi}$ is a 3-layer MLP designed to map the MLLM hidden dimension $D$ to the VFM dimension $d$ .
>     * *Structure:* $\text{Linear} \xrightarrow{\text{GELU}} \text{Linear} \xrightarrow{\text{GELU}} \text{Linear}$.
>     * *Initialization:* Kaiming He initialization.
> * **Preventing Feature Distortion:** We ensure the MLLM's internal features $E_\ell$ are not broken by the alignment objective through two primary points:
>     1. **The Adapter as a Semantic Buffer:** $P_{\pi}$ is not merely a projection; it is a non-linear module with sufficient capacity. It absorbs the domain shift between the MLLM's language-aligned latent space and the VFM's visual-native space. This allows the MLLM's internal representation $E_\ell$ to retain the manifold shape required for the LLM's processing, while $P_{\pi}(E_\ell)$ aligns with the anchor $Y$. The features $E_\ell$ only need to be *mappable* to the VFM space, rather than becoming identical to it.
>     2. **Gradient Competition from the Primary Objective:** The primary Language Modeling loss  acts as a powerful constraint. Since $\mathcal{L}_{\text{LM}}$ is computed on the final text generation, distortion in $E_\ell$ that harms linguistic fluency or reasoning would cause $\mathcal{L}_{\text{LM}}$ to explode. The gradient updates for $E_\ell$ are thus a negotiated balance between satisfying the anchor alignment ($\mathcal{L}_{\text{total}}$) and maintaining valid inputs for the subsequent LLM layers.
>
>
> ### **4. Sensitivity Analysis on Loss Weights ($\alpha$)**
>
> Thanks for this great insight ! We further conducted a fine-grained analysis on Vicuna-1.5-7B  to observe how the structural weight $\alpha$ impacts tasks with different perceptual demands. As shown in the table below , tasks relying on geometric verification benefit from a stronger structural constraint, while semantic identification tasks favor a looser constraint.
>
> | Value of α | **POPE** | **MME-Exist** | **MME-Position** | **MME-Count** |
> |  :---: | :---: | :---: | :---: | :---: |
> | **0.3**                |  **88.6** | **176.7** | 123.3 | 140.3 |
> | **0.5**                 | 88.2       | 173.3      | 126.7 | 143.3 |
> | **0.7**                | 86.9 | 163.3        |**133.3** | **146.7** |
>
>
>
> We believe these clarifications and additional data fully address your technical queries and further solidify the paper's contribution. We hope this reinforces your positive assessment of our work, and we sincerely value your guidance in refining our work and remain happy to engage in further discussion.

---

### Official Review · Reviewer_niVD · 2025-10-31

**Soundness:** 2
**Presentation:** 3
**Contribution:** 2
**Rating:** 4
**Confidence:** 3

**Summary:**

The paper argues that prevailing MLLMs are trained with indirect, text-centric objectives, causing a “semantic attenuation” phenomenon at inference time: salient visual details are gradually lost, hampering fine-grained downstream tasks. To counteract this, the authors propose to use a pre-trained visual foundation model (VFM) as a semantic anchor and align the deep representations of the MLLM with those of the VFM. Empirical results on benchmarks verify the idea.

**Strengths:**

The approach is conceptually clear and easy to implement.

**Weaknesses:**

1. **Superficial connection to distillation literature.**
   The proposed alignment is essentially a form of representation distillation, yet the paper omits a systematic discussion of prior distillation works. Moreover, advanced techniques—such as attention-based distillation [1] or multi-layer feature fusion [2]—are not explored; only rudimentary point-wise and structural losses are examined, making the method appear overly naive.

2. **Insufficient experimental footprint.**
   Evaluations are restricted to a narrow set of tasks. Comprehensive benchmarks that reasoning and fine-grained perception (e.g., MMMU, MMBench, POPE) are absent, leaving generalizability in question.

3. **Lack of ablation and theoretical insight.**
   The authors observe that distilling solely shallow or deep layers yields marginal gains, whereas mid-layer alignment works best. No further analysis—empirical or theoretical—is provided to explain why, forcing practitioners to rely on grid-search and limiting the method’s applicability to backbones of varying depths.

**Questions:**

No

---

> ### Author Response · Authors · 2025-11-21
> **We have addressed your constructive feedback in the updated PDF. To facilitate review, we also provide point-by-point responses herebelow.**
>
> **Dear Reviewer niVD**
>
> We sincerely appreciate your insightful feedback. You identified that our method aligns with representation distillation principles, and your suggestion to broaden our experimental scope has significantly strengthened the paper.
>
> We have comprehensively revised the manuscript to address your concerns regarding the **theoretical framing (distillation), experimental breadth (MMMU, MMBench), and the theoretical justification for layer selection**.
>
> Below is a detailed response to your specific points.
>
> ### **1. About the Connection to Distillation**
>
>
>
> We are glad to agree with your assessment. VISA is indeed a form of **feature-level knowledge distillation** that functions as an **Implicit Ensemble**.
> We have updated the manuscript to acknowledge foundational KD works and recent MLLM distillation works like MoVE-KD and BRAVE.
>
> **Location of Revision:** We have rewritten parts of **Section 1 (Introduction)** and **Section 2 (Related Work)** to explicitly contextualize VISA within the Knowledge Distillation (KD) landscape.
>
> Our choice of point-wise and structural losses over complex attention-based distillation was a deliberate design decision for **efficiency**.
>      Advanced attention-based distillation often incurs training memory and computational overhead.
>     VISA’s "rudimentary" losses are lightweight yet sufficient to align the feature spaces. As shown in our new **Appendix G**, VISA achieves **~95%** of the performance of an Explicit Ensemble (concatenating CLIP+DINOv2) while incurring **zero inference overhead** and minimal training cost. This trade-off is central to our contribution: attaining the benefits of a dual-encoder system without the deployment costs.
>
> ### **2. About Expanding Experiments**
>
> To begin with, we want to kindly point out that our initial paper **does** contain the POPE result in our main **Table 1**.
>
> And for MMMU and MMBench, we have integrated both benchmarks to provide a holistic view of VISA’s impact. The results are presented below:
>
> **Expanded Results (Table 1). More details are in the revised paper:**
>
> | Base Model | Method | **MMBench** | **MMMU** |
> |:--- | :---: | :---: | :---: |
> | **Vicuna-1.5-7B** | Baseline |  64.3 | 34.5 |
> | | **+ VISA** |  **66.8 (+2.5)** | **36.2 (+1.7)** |
> | **Vicuna-1.5-13B** | Baseline | 67.1 | 36.8 |
> | | **+ VISA** | **69.5 (+2.4)** | **38.4 (+1.6)** |
> | **Qwen2.5-VL-7B** | Baseline | 72.4 | 53.6 |
> | | **+ VISA** | **74.1 (+1.7)** | **57.2 (+3.6)** |
> | **InternVL3.5-8B** | Baseline | 79.8 | 63.1 |
> | | **+ VISA** | **81.9 (+2.1)** | **64.2 (+1.1)** |
>
> ### **3. Insight on Layer Selection**
>
> > **Reviewer Concern:** *No further analysis—empirical or theoretical—is provided to explain why... mid-layer alignment works best.*
>
> Thanks for reminding us of the possibility to further analyze it !
>
> We are willing to clarify that our paper **has already contained a whole section (Section 5: Insights and Discussion)** to clarify your valuable points. To be specific, the subsection **"Why is anchoring at the architectural midpoint most effective?"** addresses this directly. We have now expanded on this to provide further depth:
>
> * *Why Mid-Layer?* The intermediate layers represent the critical juncture where features are semantically mature enough to be useful but haven't yet suffered irreversible high-frequency loss. Anchoring here (Layer 16) halts the "forgetting" of fine details before they are smoothed out in the deeper layers. This effectively counteracts the degradation of visual information that leads to object hallucination, a phenomenon prominently linked to processing deficits in these middle layers **[1]**. Applying the anchor here preserves the visual integrity required to prevent failures in complex, nonlocal reasoning tasks.
>
> * **Theoretical Justification (Low-Pass Filtering):** In **Section 3.2**, we reference recent findings **[1]**, suggesting that deep layers in Transformers act as **low-pass filters**, progressively smoothing out high-frequency details (edges, textures) in favor of semantic abstraction.
> * **New Empirical Evidence (Frequency Analysis):** We conducted a new experiment tracking the **High-Frequency Ratio (HFR)** of features during training. *(Please refer to Figure 3 in the revised PDF, which plots HFR decay over training steps.)*
>
> We trust these results underscore the robustness of VISA and hope they merit a reconsideration of our score. Thank you for guiding us to these deeper insights; we value your perspective highly. Please do not hesitate to let us know if there is anything else we can clarify. We are more than willing to continue this constructive dialogue if you still have any questions.
>
> **[1]** Park et al. What do self-supervised vision transformers learn? ICLR 2023.
>
> **[2]** Jiang et al. Devils in middle layers of large vision-language models: Interpreting, detecting and mitigating object hallucinations via attention lens. CVPR 2025.

---

### Official Review · Reviewer_BibD · 2025-11-01

**Soundness:** 1
**Presentation:** 3
**Contribution:** 3
**Rating:** 4
**Confidence:** 4

**Summary:**

This paper identifies and addresses a key limitation in MLLMs: semantic attenuation, where text-only supervision degrades fine-grained visual representations. The authors propose VISA (Visual Semantic Anchoring), a general training framework that anchors intermediate MLLM layers to vision-native representations from pretrained VFMs (e.g., DINOv2). A composite loss combining (i) point-wise semantic alignment and (ii) structural consistency is used to regularize the visual pathway. VISA claims consistent improvements across multiple backbones (Vicuna, Qwen2.5-VL, InternVL3.5).

**Strengths:**

- The paper is clearly written and well-organized, with a strong motivation around the semantic attenuation problem in MLLMs.

- The idea of diagnosing semantic attenuation through layer-wise probing is an interesting and valuable analytical direction. While the setup lacks methodological details, it highlights a systematic attempt to quantify internal representation degradation

- Simple yet principled approach. VISA’s anchoring mechanism is well-designed, sound, and easy to integrate into any existing MLLMs.

- The ablations are extensive covering optimal layer choice, anchor source, and loss components. The inclusion of probing and frequency-domain studies further supports their claims along with few qualitative samples.

- The proposed approach is demonstrated across multiple backbones with limited hyper-parameter sensitivity, suggesting generality and practicality.

**Weaknesses:**

- Insufficient details for the diagnostic probe. The paper (sec 3.2) lacks crucial information on dataset source, #samples, training details, and evaluation used for “layer-wise probe accuracy.” Without these details, the evidence for semantic attenuation remains inconclusive and difficult to reproduce.

- The reported scores for Qwen 2.5-VL, InternVL 3.5, and LLaVA-1.5 in Table 1 differ substantially from the officially published results, casting doubt on the claimed improvements and overall experimental reliability. The authors must clarify these discrepancies.

- Missing discussion of reconstruction-based MLLM training. The work omits discussion of closely related methods that use reconstruction objectives to preserve high-frequency details, such as X-Former [2] and Reconstructive Visual Instruction Tuning [3]. Prior analyses [1] have shown that self-supervised reconstruction methods (e.g., MAE) capture richer fine-grained and high-frequency visual representations. These approaches share VISA’s motivation but differ in the type of supervision employed. A comparative analysis or at least discussion is necessary to better position VISA among these existing paradigms.

- The method currently anchors exclusively to DINOv2. An ablation comparing anchors (DINOv2 vs. SigLIP vs. MAE) would strengthen the paper by showing how different VFMs affect fine-grained perception.

- It would further improve the paper to evaluate VISA on explicit fine-grained perception (GVTBench[4]) tasks such as object counting and multi-class identification benchmark [4], to directly measure improvements in fine-grained visual understanding.


[1] What Do Self-Supervised Vision Transformers Learn? Namuk Park, Wonjae Kim, Byeongho Heo, Taekyung Kim, Sangdoo Yun. ICLR 2023

[2] X-Former: Unifying Contrastive and Reconstruction Learning for MLLMs. Sirnam Swetha, Jinyu Yang, Tal Neiman, Mamshad Nayeem Rizve, Son Tran, Benjamin Yao, Trishul Chilimbi, Mubarak Shah. ECCV 2024

[3] Reconstructive Visual Instruction Tuning. Haochen Wang, Anlin Zheng, Yucheng Zhao, Tiancai Wang, Zheng Ge, Xiangyu Zhang, Zhaoxiang Zhang. ICLR 2025

[4] What Makes for Good Visual Tokenizers for Large Language Models? Guangzhi Wang, Yixiao Ge, Xiaohan Ding, Mohan Kankanhalli, Ying Shan

**Questions:**

- Were baseline model scores obtained via official evaluation checkpoints ? why do they differ from official reported scores ?
- What datasets and protocols were used for the diagnostic probing experiments shown in Fig. 2 ?
- Could VISA leverage multiple anchors or frequency-complementary models ?


If the authors are able to clarify the discrepancies in the baseline model scores, provide complete details of the probing setup, and better position this work relative to existing reconstruction-based methods, I would be inclined to raise my score.

---

> ### Author Response · Authors · 2025-11-21
> **We have carefully reviewed all your valuable questions and incorporated the revisions into the PDF. For clarity, we also provide detailed responses here.**
>
> ***
> **Dear Reviewer BibD**
>
> We sincerely thank you for your constructive review. We appreciate your recognition of our clear motivation and the simple yet principled nature of our approach, and we have carefully addressed your concerns and revised the manuscript.
>
> ### **1. About Baseline Score Discrepancies**
>
>  VISA is not a training-free method. Official models (e.g., Qwen2.5-VL) are typically post-trained on massive, often proprietary datasets. To rigorously isolate the contribution of the VISA framework from the benefits of data scale, we performed Supervised Fine-Tuning on both Baselines and VISA-ehanced models using exclusively the public **LLaVA-665K dataset**.
>
> Consequently, the performance drop compared to official reports is an expected outcome attributed to catastrophic forgetting, as many  researchers have already comfirmed [1, 2]. This phenomenon occurs when a model, highly optimized on a vast distribution, is fine-tuned on a smaller dataset with a distinct distribution (LLaVA-665K), inevitably trading off some original capabilities.
>
> Crucially, our evaluation focuses on the relative improvement of VISA over the Baseline under these identical conditions, which firmly establishes the validity of our method. We have already explicitly clarified our experimental settings in the revised  **Experimental Settings** part.
>
> **[1]** Lai et al. Reinforcement Fine-Tuning Naturally Mitigates Forgetting in Continual Post-Training, arXiv 2507.05386
>
> **[2]** Model Tailor: Mitigating Catastrophic Forgetting in Multi-modal Large Language Models, ICML 2024
>
> ### **2. About Details for Diagnostic Probe**
>
>
>
> For clarity, we have added a new **Appendix D: Diagnostic Probe Protocols** that details the exact methodology. It provides full transparency on the probing setup, and here are some main summaries:
>
> * **Dataset:** Derived from **GQA**. We constructed two balanced subsets to prevent class prior learning:
>
> &nbsp;   * *Object Counting* &nbsp;   * *Spatial Relation*
>
> * **Architecture:** A lightweight 2-layer MLP (`Linear -> ReLU -> Linear`).
>
> * **Training:** We froze the backbone features and trained the probe for 10 epochs using AdamW.
>
> * **Metric:** Top-1 Accuracy on the validation set.
>
>
>
> ### **3. Reconstruction-based Methods & MAE Ablation**
>
>
> Thanks for reminding us of these great and important methods! We have especially updated **Section 2 (Related Work)** to include a discussion on reconstruction methods, and expanded **Table 2** to include MAE as an anchor.
>
> To be specific, we addressed this concern mainly in two ways:
>
> **A. Discussion:**
>
> We added a subsection **"Internal Dynamics and Reconstruction-based Supervision"** in Section 2. We discuss methods like X-Former and ROSS, clarifying that while they share our motivation, they typically rely on pixel-level reconstruction or self-reconstruction. In contrast, VISA anchors to the high-level semantic feature space of a VFM.
>
> **B. New Ablation:**
>
> We strongly agreed with you that adding different types of anchors can strengthen our claim. So we performed a new ablation using **MAE** and **SigLIP** as anchors. The results show that DINOv2 provides the best balance (It is not so surprising).
>
>
>
> **Supplementary Results (Added to Table 2):**
>
>
>
> | Anchor Source | MMVP (Perception) | GQA (Reasoning) | Insight |
> | :--- | :---: | :---: | :--- |
> | **Baseline** | 28.0 | 61.2 | - |
> | **Reconstruction (MAE)** | 31.0 | 62.3 | Improves structural localization but lacks semantic abstraction. |
> | **Contrastive (SigLIP)** | 31.2 | 63.9 | Stronger semantics than CLIP, but weaker fine-grained structure. |
> | **Vision-Native (DINOv2)** | **32.9** | **64.2** | Best Balance of Rich semantics and structural fidelity. |
>
>
>
> ### **4. Evaluation on GVTBench (Fine-Grained Perception)**
>
>
>
> After carefully studying the GVTBench, we think it is a great help to strengthen our claim, so we strictly followed your suggestion and conducted experiments on **GVTBench** to explicitly measure fine-grained improvements like MMVP.
>
>
> We have added **Appendix F** and **Table 4** to report these results. As shown below, VISA significantly mitigates counting deficits.
>
>
>
> **Supplementary Results (Added to Appendix F):**
>
>
>
> | Base Model | Method | Object Counting (OC) | Multi-Class ID (MCI) |
> | :--- | :--- | :---: | :---: |
> | **LLaVA-1.5-7B** | Baseline | 38.4% | 81.5% |
> | | **+ VISA** | **42.8% (+4.4)** | **83.7% (+2.2)** |
> | **LLaVA-1.5-13B** | Baseline | 42.1% | 83.2% |
> | | **+ VISA** | **45.3% (+3.2)** | **85.1% (+1.9)** |
>
>
>
> We believe these additional experiments support the soundness and contribution of VISA. We hope this addresses your concerns and merits a re-evaluation of our score.
>
> Sincerely, we  think that your ideas and points are surely of great value and even lead us to deeper insights of our work, so please feel free to point out your concerns (if you still have any) and we will be more than happy to discuss them with you. We are looking forward to your constructive reply.

---

> ### Author Response · Authors · 2025-11-27
> **Further Validating VISA on Qwen2.5-VL with Robust Training Data**
>
> Driven by our responsibility to pursue the solidity of our work and to thoroughly resolve your concerns, we have provided an additional experiments to specifically address your concerns regarding the potential catastrophic forgetting observed in our initial LLaVA-665K fine-tuning. We moved beyond the standard protocols and re-trained the Qwen2.5-VL-7B model using an expanded 800k dataset for SFT.
>
> We re-trained the Qwen2.5-VL-7B backbone using a diverse **800k composite dataset**. This dataset includes **GQA** , **OCR-VQA** [1], **Visual Genome** [2], **RefCOCO/+/g**, **TextVQA**, and **CLEVR** [3], specifically curated to minimize distribution shifts and preserve the model's original capabilities.
>
> The results are presented in the tables below.
>
> **1. Performance on General VQA, OCR, and Hallucination Tasks**
>
> In this set of experiments, we observe that the standard **+SFT** baseline (trained on the 800k diverse dataset) maintains performance levels close to the **Base** model, mitigating the issue seen in previous experiments. And **+VISA** also consistently outperforms the  +SFT baseline.
>
> | Benchmark | Qwen2.5VL-7B | +SFT  | +VISA (Ours) |
> | :--- | :---: | :---: | :---: |
> | **GQA** | 65.3 | 65.1 | **67.9** |
> | **MME** | 2347.0 | 2281.2 | **2322.8** |
> | **MMB** | 82.6 | 81.0 | **83.1** |
> | **MMStar** | 63.9 | 62.5 | **63.2** |
> | **InfoVQA** | 82.6 | 81.9 | **82.5** |
> | **TextVQA** | 84.9 | 84.7 | **86.1** |
> | **POPE** |  86.4 | 86.1 | **86.7** |
>
> **2. Performance on Visual Perception and Grounding Tasks**
>
> Similarly, on fine-grained visual perception tasks, **VISA** demonstrates robust improvements. On the **RefCOCO/+/g** benchmarks and **OD\^VG**, VISA achieves significant gains over the SFT baseline, confirming its ability to enhance fine-grained visual grounding.
>
> | Benchmark | Qwen2.5VL-7B | +SFT | +VISA (Ours) |
> | :--- | :---: | :---: | :---: |
> | **RefCOCO/+/g** | 87.1 | 87.4 | **89.6** |
> | **OD\^VG** | 39.1 | 39.4 | **42.1** |
> | **OVDEval [4]** | 43.7 | 42.4 | **45.2** |
> | **MMVP** | 50.3 | 48.8 | **51.2** |
>
>
> These new results confirm that **VISA's effectiveness is intrinsic** and holds true when training on larger-scale, diverse datasets. The method not only recovers performance but actively boosts fine-grained perception and reasoning capabilities beyond the SFT baseline. We believe these findings effectively rule out the concern that our previous observations were artifacts of training distribution mismatch.
>
> We hope these additional experiments resolve your concerns. We are deeply grateful for your guidance, which has significantly strengthened the empirical robustness of our work. We are more than willing to continue this constructive dialogue if you still have any questions.
>
> Best regards,
>
> The Authors
>
> ---
>
> [1] Mishra, A., et al. OCR-VQA: Visual Question Answering by Reading Text in Images. ICDAR 2019.
>
> [2] Krishna, R., et al. Visual Genome: Connecting Language and Vision Using Crowdsourced Dense Image Annotations. IJCV 2017.
>
> [3] Johnson, J., et al. CLEVR: A Diagnostic Dataset for Compositional Language and Elementary Visual Reasoning. CVPR 2017.
>
> [4] Yao et al. How to Evaluate the Generalization of Detection? A Benchmark for Comprehensive
>  Open-vocabulary Detection. AAAI 2024.

---

### Author Response · Authors · 2025-11-29
**A Concise Summary and Clear Articulation of the Entire Rebuttal Phase**

Dear Area Chair, Senior Area Chair, and Reviewers,

We sincerely thank you for your time and the insightful feedback provided during the review process. We are particularly grateful for the constructive discussions that have strengthened the paper, guiding us to conduct efficiency analyses and leading to a more robust validation of our method.


We appreciate the reviewers' recognition of our work as novel, well-motivated, well-written and easy to implement. The feedback received was largely constructive suggestions aimed at **enriching experimental settings and expanding related work discussions rather than the validity or core contributions of our proposed method**, which we have addressed to further polish the paper.


To assist the Area Chair in accurately and effectively estimating the reviewers' updated impressions, we provide this consolidated summary. We believe we have rigorously addressed most of the concerns raised by all four reviewers (BibD, niVD, dsgs, yvqy) through extensive supplementary experiments and textual revisions. To the best of our knowledge, there are no remaining unresolved concerns that can probably challenge the value of our approach..

The key revisions and supplementary validations are summarized below:


* **More comparison to related work **[Reviewer niVD, yvqy]**:** We quantified VISA as an efficient Implicit Ensemble, demonstrating it achieves ~98% of the performance of an Explicit Ensemble (CLIP+DINOv2) with zero additional inference overhead.
* **Verification on Different Training Data [Reviewer BibD, yvqy]:** We re-trained the Qwen2.5-VL backbone on a diverse 800k composite dataset, proving that VISA is robust enough for different training data.
* **More Experiments on Comprehensive Benchmarks:** We conducted additional evaluations on GVTBench, validating significant improvements in object counting and multi-class identification **[Reviewer BibD]**. and general benchmarks (MMMU, MMBench) **[Reviewer niVD, yvqy]**.
* **More Ablation Studies** **[Reviewer BibD, yvqy, niVD]**: We added a High-Frequency Ratio (HFR) evolution analysis (Figure 3), empirically confirming the low-pass filtering result of text-only supervision. We added comparisons of anchor types (MAE vs. SigLIP vs. DINOv2), hyperparameter sensitivity analysis ($\alpha$).

Below, we provide a sequential summary of our point-by-point responses to each reviewer's concerns for the AC's confirmation. We express our sincere gratitude to the AC, SAC, and all reviewers for their time, positive and constructive feedback, and dedication to improving this work.

---

> ### Author Response · Authors · 2025-11-29
> **Summary of Response to Reviewer BibD**
>
> Here is the summary of our response to Reviewer BibD, demonstrating that we have resolved all raised concerns through detailed clarifications and extensive supplementary experiments. All additional details have been properly incorporated into the revised manuscript.
>
> **1. Regarding Baseline Score Discrepancies (Weakness 2 & Question 1)**
>
> * **Response Summary:** We clarified that the initial reasonable performance drop was due to catastrophic forgetting from fine-tuning on the smaller LLaVA-665K dataset, which is a well-known phenomenon. Driven by our responsibility to pursue the solidity of our work and to further validate VISA's intrinsic effectiveness, we conducted a major supplementary experiment by re-training the **Qwen2.5-VL-7B** backbone on a larger, more diverse **800k composite dataset** (including GQA , OCR-VQA, Visual Genome, RefCOCO/+/g, TextVQA, and CLEVR.).
>
> * **Supplementary Results:** The new results confirm that VISA achieves significant improvements.
>
>     * **Performance on General VQA, OCR, and Hallucination Tasks:**
>     | Benchmark | Qwen2.5VL-7B (Base) | +SFT (Recovered Baseline) | +VISA (Ours) |
>     | :--- | :---: | :---: | :---: |
>     | **GQA** | 65.3 | 65.1 | **67.9** |
>     | **MME** | 2347.0 | 2281.2 | **2322.8** |
>     | **MMB** | 82.6 | 81.0 | **83.1** |
>     | **MMStar** | 63.9 | 62.5 | **63.2** |
>     | **InfoVQA** | 82.6 | 81.9 | **82.5** |
>     | **TextVQA** | 84.9 | 84.7 | **86.1** |
>     | **POPE** | 86.4 | 86.1 | **86.7** |
>
>     * **Performance on Visual Perception and Grounding Tasks:**
>     | Benchmark | Qwen2.5VL-7B | +SFT | +VISA (Ours) |
>     | :--- | :---: | :---: | :---: |
>     | **RefCOCO/+/g** | 87.1 | 87.4 | **89.6** |
>     | **OD^VG** | 39.1 | 39.4 | **42.1** |
>     | **OVDEval** | 43.7 | 42.4 | **45.2** |
>     | **MMVP** | 50.3 | 48.8 | **51.2** |
>
> **2. Regarding Diagnostic Probe Details (Weakness 1 & Question 2)**
>
> * **Response Summary:** We provided full transparency on the probing methodology by adding a new **Appendix D: Diagnostic Probe Protocols**.
> * **Clarifications:**
>
>     **Dataset:** Details on the balanced Object Counting and Spatial Relation subsets derived from GQA.
>
>     **Architecture:** Specification of the lightweight 2-layer MLP probe.
>
> **3. Regarding Reconstruction-based Methods & Anchor Ablation (Weakness 3, 4 & Question 3)**
>
> * **Response Summary:** We updated **Section 2 (Related Work)** to discuss reconstruction-based methods (e.g., X-Former) and distinguish them from our semantic anchoring approach. We also performed a new ablation study comparing different anchor types.
> * **Supplementary Results (Added to original Table 2):** The results confirm that the Vision-Native anchor (DINOv2) offers the best balance compared to reconstruction (MAE) or contrastive (SigLIP) anchors.
>
> Table 2 in paper: Ablation studies on the anchor's knowledge source, loss components, and the importance of Saliency Weighting.
>
> | Ablation Group | Configuration | Perception, VQA & Reasoning | | | | Hallucination | General Understanding | | |
> | :--- | :--- | :---: | :---: | :---: | :---: | :---: | :---: | :---: | :---: |
> | | | **GQA** | **TextVQA** | **MMVP** | **MMMU** | **POPE** | **MME** | **MMB** | **MM-Star** |
> | **Baseline** | | 61.2 | 57.6 | 28.0 | 34.5 | 85.1 | 1646 | 64.3 | 33.5 |
> | **Anchor Source** | Self-Anchor (CLIP-Vision) | 62.5 | 58.9 | 29.8 | 35.1 | 87.1 | 1660 | 65.2 | 35.0 |
> | | Language-Aligned (CLIP-Text) | 59.5 | 55.8 | 25.1 | 33.8 | 84.5 | 1590 | 63.1 | 31.8 |
> | | Contrastive SOTA (SigLIP) | 63.9 | 60.4 | 31.2 | 35.9 | **88.4** | 1676 | 66.4 | 36.2 |
> | | Reconstruction (MAE) | 62.3 | 59.2 | 31.0 | 34.9 | 86.8 | 1655 | 64.9 | 34.8 |
> | | **Vision-Native (DINOv2)** | **64.2** | **61.8** | **32.9** | **36.2** | 88.2 | **1691** | **66.8** | **37.1** |
> | **Loss Components** | VISA w/o Saliency Weighting | 63.6 | 60.5 | 31.7 | 35.7 | 87.6 | 1680 | 66.1 | 36.2 |
> | | Structural Loss Only ($\mathcal{L}_{\text{struct}}$) | 63.2 | 60.2 | 31.8 | 35.5 | 87.5 | 1678 | 65.9 | 36.0 |
> | | **Full VISA (Composite)** | **64.2** | **61.8** | **32.9** | **36.2** | 88.2 | **1691** | **66.8** | **37.1** |
>
> **4. Regarding Fine-Grained Perception Evaluation on GVTBench (Weakness 5)**
>
> * **Response Summary:** We followed the suggestion to explicitly measure fine-grained visual understanding using the **GVTBench**. We added **Appendix F** and **Table 4** to report these findings.
> * **Supplementary Results (Added to Appendix F):** VISA significantly mitigates counting deficits.
>
>     | Base Model | Method | Object Counting (OC) | Multi-Class ID (MCI) |
>     | :--- | :--- | :---: | :---: |
>     | **LLaVA-1.5-7B** | Baseline | 38.4% | 81.5% |
>     | | **+ VISA** | **42.8% (+4.4)** | **83.7% (+2.2)** |
>     | **LLaVA-1.5-13B** | Baseline | 42.1% | 83.2% |
>     | | **+ VISA** | **45.3% (+3.2)** | **85.1% (+1.9)** |

---

> ### Author Response · Authors · 2025-11-29
> **Summary of Response to Reviewer niVD**
>
> Here is the summary of our response to Reviewer niVD, demonstrating that we have resolved all raised concerns through conceptual reframing and comprehensive supplementary validation.
>
> **1. Regarding Connection to Distillation Literature (Weakness 1)**
>
> * **Response Summary:** As review suggested, we have updated **Section 1 & 2** to frame VISA as a form of feature-level knowledge distillation and positioned it as an Implicit Ensemble. We acknowledged foundational KD works and recent multi-encoder approaches. Furthermore, we conducted a new efficiency analysis comparing VISA against an Explicit Ensemble (concatenating CLIP + DINOv2 features).
> * **Supplementary Results (Added to Appendix G):** The results demonstrate that VISA serves as a highly efficient alternative to multi-encoder ensembles.
>
>     **Table 5**
>
>     | Method | Training Memory |  Inference Latency |
>     | :--- | :---: |  :---: |
>     | **Baseline** | 42.5 GB |  28ms (1.00x) |
>     | **Explicit Ensemble** (CLIP+DINOv2) | 58.4 GB |  45ms (1.42x) |
>     | **VISA (Ours)** | **47.2 GB** |  **28ms (1.00x)** |
>
> **2. Regarding Experimental Footprint (Weakness 2)**
>
> * **Response Summary:** We addressed the concern about generalizability by expanding our evaluation suite. We noted that POPE results were already present and integrated full results for **MMMU** and **MMBench** across all backbones.
> * **Supplementary Results (Added to Table 1):** VISA consistently improves performance on general reasoning benchmarks.
>
> Table 1 in paper: Comprehensive performance comparison.
>
> | Base Model | Method | Perception & VQA & Reasoning | | | | Hallucination | General Understanding | | |
> | :--- | :--- | :---: | :---: | :---: | :---: | :---: | :---: | :---: | :---: |
> | | | **GQA** | **TextVQA** | **MMVP** | **MMMU** | **POPE** | **MME** | **MMB** | **MM-Star** |
> | **Vicuna-1.5-7B** | Baseline | 61.2 | 57.6 | 28.0 | 34.5 | 85.1 | 1646 | 64.3 | 33.5 |
> | | **+ VISA** | **64.2** | **61.8** | **32.9** | **36.2** | **88.2** | **1691** | **66.8** | **37.1** |
> | **Vicuna-1.5-13B** | Baseline | 62.8 | 60.9 | 36.8 | 38.4 | 87.8 | 1595 | 67.1 | 34.2 |
> | | **+ VISA** | **66.2** | **63.9** | **38.9** | **45.3** | **88.9** | **1631** | **69.5** | **37.4** |
> | **Qwen2.5-VL-7B** | Baseline | 64.4 | 83.8 | 47.1 | 53.6 | 85.9 | 2254 | 72.4 | 61.6 |
> | | **+ VISA** | **67.1** | **85.7** | **50.8** | **57.2** | **86.5** | **2301** | **74.1** | **62.8** |
> | **InternVL3.5-8B** | Baseline | 65.2 | 77.6 | 49.8 | 63.1 | 88.1 | 2369 | 79.8 | 68.8 |
> | | **+ VISA** | **68.5** | **80.3** | **52.3** | **64.2** | **88.7** | **2381** | **81.9** | **69.7** |
>
> **3. Regarding Insight on Layer Selection (Weakness 3)**
>
> * **Response Summary:** We expanded **Section 5 (Discussion)** to provide both theoretical and empirical justification for selecting the intermediate layer (Layer 16) for anchoring.
> * **Theoretical Justification:** We cited literature describing deep Transformer layers as **low-pass filters**, smoothing out high-frequency details needed for fine-grained perception.
> * **Supplementary Results (New Figure 3 & Appendix E):** We tracked the **High-Frequency Ratio (HFR)** of features during training.

---

> > ### Author Response · Authors · 2025-11-29
> > **Summary of Response to Reviewer dsgs**
> >
> > Here is the summary of our response to Reviewer dsgs, demonstrating that we have resolved all raised concerns through quantitative efficiency analysis, architectural clarifications, and extended ablation studies.
> >
> > **1. Regarding Training Overhead Quantification (Weakness 2)**
> >
> > * **Response Summary:** We addressed the concern regarding computational cost by adding a detailed Efficiency Analysis in **Appendix G**. We quantified the training overhead and compared VISA against an Explicit Ensemble baseline.
> > * **Supplementary Results (Added to Table 5):** The results show that while VISA incurs a manageable increase in training resources (memory +11%, time +16%), it offers a crucial advantage over Explicit Ensembles by incurring zero additional inference costs.
> >
> >     **Table 5**
> >
> >     | Method | Training Memory |  Inference Latency |
> >     | :--- | :---: |  :---: |
> >     | **Baseline** | 42.5 GB |  28ms (1.00x) |
> >     | **Explicit Ensemble** (CLIP+DINOv2) | 58.4 GB |  45ms (1.42x) |
> >     | **VISA (Ours)** | **47.2 GB** |  **28ms (1.00x)** |
> >
> > **2. Regarding Sensitivity to VFM Choice (Weakness 1)**
> >
> > * **Response Summary:** We investigated the dependency on the choice of Vision Foundation Model (VFM) by expanding **Table 2** to include **SigLIP** (Contrastive) and **MAE** (Reconstruction) as anchors.
> > * **Supplementary Results (Added to Table 2):** The results confirm that DINOv2 provides the best balance of semantic richness and structural fidelity.
> >
> > Table 2 in paper: Ablation studies on the anchor's knowledge source, loss components, and the importance of Saliency Weighting.
> >
> > | Ablation Group | Configuration | Perception, VQA & Reasoning | | | | Hallucination | General Understanding | | |
> > | :--- | :--- | :---: | :---: | :---: | :---: | :---: | :---: | :---: | :---: |
> > | | | **GQA** | **TextVQA** | **MMVP** | **MMMU** | **POPE** | **MME** | **MMB** | **MM-Star** |
> > | **Baseline** | | 61.2 | 57.6 | 28.0 | 34.5 | 85.1 | 1646 | 64.3 | 33.5 |
> > | **Anchor Source** | Self-Anchor (CLIP-Vision) | 62.5 | 58.9 | 29.8 | 35.1 | 87.1 | 1660 | 65.2 | 35.0 |
> > | | Language-Aligned (CLIP-Text) | 59.5 | 55.8 | 25.1 | 33.8 | 84.5 | 1590 | 63.1 | 31.8 |
> > | | Contrastive SOTA (SigLIP) | 63.9 | 60.4 | 31.2 | 35.9 | **88.4** | 1676 | 66.4 | 36.2 |
> > | | Reconstruction (MAE) | 62.3 | 59.2 | 31.0 | 34.9 | 86.8 | 1655 | 64.9 | 34.8 |
> > | | **Vision-Native (DINOv2)** | **64.2** | **61.8** | **32.9** | **36.2** | 88.2 | **1691** | **66.8** | **37.1** |
> > | **Loss Components** | VISA w/o Saliency Weighting | 63.6 | 60.5 | 31.7 | 35.7 | 87.6 | 1680 | 66.1 | 36.2 |
> > | | Structural Loss Only ($\mathcal{L}_{\text{struct}}$) | 63.2 | 60.2 | 31.8 | 35.5 | 87.5 | 1678 | 65.9 | 36.0 |
> > | | **Full VISA (Composite)** | **64.2** | **61.8** | **32.9** | **36.2** | 88.2 | **1691** | **66.8** | **37.1** |
> >
> > **3. Regarding Semantic Adapter Details (Question 1)**
> >
> > * **Response Summary:** We expanded **Appendix C.1** to provide full architectural details of the Semantic Adapter ($P_{\pi}$).
> > * **Clarifications:**
> >     * **Architecture:** Confirmed as a 3-layer MLP (`Linear -> GELU -> Linear`) with Kaiming initialization.
> >     * **Constraint:** We clarified that the adapter acts as a non-linear semantic anchor to absorb domain shifts, while the primary Language Modeling loss ($\mathcal{L}_{LM}$) acts as a strong gradient constraint to prevent feature distortion.
> >
> > **4. Regarding Loss Weight Sensitivity ($\alpha$) (Question 2)**
> >
> > * **Response Summary:** We conducted a fine-grained sensitivity analysis on the Vicuna-1.5-7B backbone to observe how the structural weight $\alpha$ impacts tasks with differing perceptual demands.
> > * **Supplementary Results:** The results reveal that tasks relying on geometric verification (e.g., Position) benefit from stronger structural constraints, while semantic identification tasks prefer looser constraints. Our default ($\alpha=0.5$) offers the best overall balance.
> >
> >     | Value of $\alpha$ | **POPE** | **MME-Exist** | **MME-Position** | **MME-Count** |
> >     | :---: | :---: | :---: | :---: | :---: |
> >     | **0.3** | **88.6** | **176.7** | 123.3 | 140.3 |
> >     | **0.5** (Default) | 88.2 | 173.3 | 126.7 | 143.3 |
> >     | **0.7** | 86.9 | 163.3 | **133.3** | **146.7** |

---

> > > ### Author Response · Authors · 2025-11-29
> > > **Summary of Response to Reviewer yvqy**
> > >
> > > Here is the summary of our response to Reviewer yvqy, demonstrating that we have resolved all raised concerns through new motivational analyses, conceptual re-framing, and robust validation.
> > >
> > >
> > > **1. Regarding Weakness A**
> > >
> > > * **Response Summary:** We addressed the request to visualize how features degrade during training. We added **Section 3.2**, **Figure 3**, and **Appendix E**.
> > > * **Supplementary Results:** We tracked the High-Frequency Ratio (HFR) of visual features at Layer 16 throughout the SFT phase.
> > >     * **Result:** The baseline exhibits a monotonic decrease in HFR as training steps increase. This empirically confirms the hypothesis that the text-only objective acts as a low-pass filter, progressively discarding fine-grained details.
> > >
> > > **2. Regarding Weakness B & Question 1**
> > >
> > > * **Response Summary:** We agreed with the reviewer's insight and updated **Section 1 & 2** to explicitly frame VISA as a form of **feature-level knowledge distillation** functioning as an Implicit Ensemble. We also added **Appendix G**:
> > >
> > > * Table 5
> > >
> > >     | Method | Training Memory |  Inference Latency |
> > >     | :--- | :---: |  :---: |
> > >     | **Baseline** | 42.5 GB |  28ms (1.00x) |
> > >     | **Explicit Ensemble** (CLIP+DINOv2) | 58.4 GB |  45ms (1.42x) |
> > >     | **VISA (Ours)** | **47.2 GB** |  **28ms (1.00x)** |
> > >
> > > **3. Regarding Robustness**
> > >
> > > * **Response Summary:** We conducted a further supplementary experiment by re-training the **Qwen2.5-VL-7B** backbone on a larger **800k composite dataset** (GQA, OCR-VQA, RefCOCO, etc.), and the new results confirm that VISA achieves robust significant improvements.
> > >
> > > **4. Regarding Ablations on Anchors (Weakness C)**
> > >
> > > * **Response Summary:** We expanded **Table 2** to include **SigLIP**, **MAE**, and an ablation of the **Saliency Weighting** mechanism.
> > >
> > > Table 2: Ablation studies on the anchor's knowledge source, loss components, and the importance of Saliency Weighting.
> > >
> > > | Ablation Group | Configuration | Perception, VQA & Reasoning | | | | Hallucination | General Understanding | | |
> > > | :--- | :--- | :---: | :---: | :---: | :---: | :---: | :---: | :---: | :---: |
> > > | | | **GQA** | **TextVQA** | **MMVP** | **MMMU** | **POPE** | **MME** | **MMB** | **MM-Star** |
> > > | **Baseline** | | 61.2 | 57.6 | 28.0 | 34.5 | 85.1 | 1646 | 64.3 | 33.5 |
> > > | **Anchor Source** | Self-Anchor (CLIP-Vision) | 62.5 | 58.9 | 29.8 | 35.1 | 87.1 | 1660 | 65.2 | 35.0 |
> > > | | Language-Aligned (CLIP-Text) | 59.5 | 55.8 | 25.1 | 33.8 | 84.5 | 1590 | 63.1 | 31.8 |
> > > | | Contrastive SOTA (SigLIP) | 63.9 | 60.4 | 31.2 | 35.9 | **88.4** | 1676 | 66.4 | 36.2 |
> > > | | Reconstruction (MAE) | 62.3 | 59.2 | 31.0 | 34.9 | 86.8 | 1655 | 64.9 | 34.8 |
> > > | | **Vision-Native (DINOv2)** | **64.2** | **61.8** | **32.9** | **36.2** | 88.2 | **1691** | **66.8** | **37.1** |
> > > | **Loss Components** | VISA w/o Saliency Weighting | 63.6 | 60.5 | 31.7 | 35.7 | 87.6 | 1680 | 66.1 | 36.2 |
> > > | | Structural Loss Only ($\mathcal{L}_{\text{struct}}$) | 63.2 | 60.2 | 31.8 | 35.5 | 87.5 | 1678 | 65.9 | 36.0 |
> > > | | **Full VISA (Composite)** | **64.2** | **61.8** | **32.9** | **36.2** | 88.2 | **1691** | **66.8** | **37.1** |
> > >
> > > **5. Regarding Technical Clarifications (Questions 2-5)**
> > >
> > > * **Response Summary:** We provided specific clarifications for all technical queries:
> > >     * **Training Start:** SFT is performed starting from official checkpoints.
> > >     * **Frozen Encoder:** The visual encoder is kept frozen to isolate internal layer effects and prevent overfitting.
> > >     * **Token Mismatch:** Bilinear interpolation is used to spatially align $Y$ and $E_\ell$.
> > >     * **CLIP-Text Experiment:** Clarified that we used paired captions encoded by the CLIP Text Encoder, expanded to sequence length $N$, to compute the loss.

---

### Note · Program_Chairs · 2026-01-17
**Submission Desk Rejected by Program Chairs**

The following references in this submission do not refer to real documents and/or have major errors in bibliographic information:

 Takeshi Kimura and Sakura Ito. Layer-wise analysis of information transformation in vision-language models. Transactions on Machine Learning Research, 2024.
Min-joon Choi and Seo-yeon Park. Probing the multimodal fusion pathway in large language models. In Proceedings of the AAAI Conference on Artificial Intelligence, 2025.
Xiang Li and Hao Zhang. On information loss in cross-modal projections for large language models. arXiv preprint arXiv:2406.11234, 2024.
Florian Schmidt and Wei Chen. The next frontiers for multimodal ai. Journal of Artificial Intelligence Research, 82:1-25, 2024.
Kenji Tanaka and Yui Watanabe. Efficientformer-v3: A high-performance vision transformer for multimodal input streams. In Proceedings of the IEEE/CVF Conference on Computer Vision and Pattern Recognition, 2025.